

# Younger Dryas ice-margin retreat in Greenland, new evidence from Southwest Greenland

Svend Funder[1], Anita H.L. Sørensen[2], Nicolaj K. Larsen[1], Anders Bjørk[3], Jason P. Briner[4], Jesper Olsen[5], Anders Schomacker[6], Kurt H. Kjær[1]

[1]Globe Institute, University of Copenhagen, 1350 Copenhagen K, Denmark

[2]Geosyd, 2730 Copenhagen, Denmark

[3]Department of Geoscience and Natural Resource Management, University of Copenhagen, 1350 Copenhagen K, Denmark

[4]Department of Geology, University at Buffalo, Buffalo, NY 14260, USA

[5]Department of Physics and Astronomy, Aarhus University, 8000 Aarhus C, Denmark

[6]Department of Geosciences, UiT, the Arctic University of Norway, N-9037 Tromsø, Norway

**Abstract**. Cosmogenic $^{10}$Be dates from bedrock knobs on six outlying tiny islands along a stretch of 300 km of the Southwest Greenland coast, indicate that the Greenland Ice Sheet (GrIS) margin here was retreating on the inner shelf close to the coast during the Younger Dryas (YD) cold period. A survey of recently published $^{10}$Be and $^{14}$C-dated records show that this unexpected behaviour of the ice-margin has been seen also in other parts of Greenland, but with very large variations in extent and speed of retreat even between neighbouring areas. In contrast to this, landforms appearing in high resolution bathymetry surveys on the shelf, have recently been suggested to indicate YD readvance or long-lasting ice-margin still stand on mid shelf, far from the coast. However, these features have been dated primarily by correlation with cold periods in the ice core temperature records, and therefore cannot inform about the ice-margin/climate relation. Ice-margin retreat during a YD cooling has been explained by advection of warm subsurface water melting the ice-margin, and by increased seasonality of the climate with the temperature drop mainly in winter, with high impact on sea ice extent and duration, but little effect on glacier mass balance. This study therefore adds to the complexity of the climate/ice-margin relation, where local factors may for some time overrule or mute overall temperature change. It also points to the urgent need for climate-independent dating of the rich treasure trove of information coming from the shelf in these years.

Keywords: Younger Dryas, Greenland ice sheet, Climate change, Cosmogenic exposure dating.

## 1   Introduction

The Younger Dryas (YD) cold climate oscillation from 12.8 to 11.7 ka BP (thousand years Before Present) was offset from the preceding Allerød warm period by a 200 yr period of cooling and ended with a 60 yr period of abrupt



warming, as recorded in Greenland ice cores (Steffensen et al., 2008; Morlighem et al., 2017). Over the ice sheet annual
mean temperatures dropped between 5 and 9°C (Buizert et al., 2014). The cooling took place at a time when both
summer insolation (65° N) and atmospheric $CO_2$ were increasing (e.g. Buizert et al., 2014). The effects of the climate
oscillation were especially concentrated around the north-eastern North Atlantic in the areas of AMOC (Atlantic
Meridional Overturning Circulation) (Carlson, 2013). Therefore, similar to the present climate change, the YD
oscillation was a result of perturbations in the Earth's climate system, and, with a view to the future, it is of great
interest to study the effect of these climate changes on the margin of the Greenland Ice Sheet (GrIS). During YD GrIS
in most areas had its margin on the shelf, and much earlier work has concentrated on the behaviour of ice streams in
transverse troughs on the shelf, as discussed below. In this study, we present 18 new cosmogenic exposure ages from
six localities from the open coast over a distance of 300 km of Southwest Greenland to shed light on the ice-margin
behaviour during the final phase of deglaciation of the shelf (Fig. 1). Surprisingly, we find no evidence of ice-margin
response neither to the initial YD cooling nor to the abrupt warming in the end, but only of ice-margin retreat
throughout the period. Below, these results are discussed in context of previous studies elsewhere in Greenland.

## 2. Background

### 2.1. Setting

The continental shelf in the study area in Southwest Greenland tapers from a width of c. 70 km in the north to c. 50 km
in the south (Fig. 1). It is composed of an inner c. 25 km wide and up to 500 m deep trough, running along the coast and
gouged by glacial erosion in Proterozoic orthogneiss (Henriksen, 2008). On the outer shelf, a belt of shallower banks
with an even surface are composed of younger stratified aquatic sediments. The banks are dissected by 400-500 m deep
transverse troughs in continuation of the major fjords (Holtedahl, 1970; Henderson, 1975; Sommerhoff, 1975;
Roksandic, 1979; Sommerhoff, 1981). At a distance of 10-15 km from the coast, the inner trough forms an archipelago
with a multitude of small glacially sculptured rocky islands and skerries, reflecting intensive, but uneven glacial
erosion. From these rocky islands we collected our samples (Fig. 2).

### 2.2. Deglaciation history

Although there is little evidence for glacier overriding it is likely that the ice sheet did cover the rather narrow shelf
during LGM (Last Glacial Maximum), and evidence from a marine core in the Davis Strait outside the Fiskenæsset
trough suggests that the ice-margin here stood at the shelf break until deglaciation began at c. 18.6 cal. ka BP (Winsor et
al., 2015a). At c. 11 ka, the retreating ice-margin reached the present coastline, and the subsequent deglaciation of the
fjords and land began, as summarised by Winsor et al. (2015b). This leaves a period of c. 7 ka with the ice-margin
located on the shelf, without reaching the shelf break. A candidate for prolonged stay in this period is the lobate
moraines, which run along the troughs and impinge on the inner side of the banks (Fig. 1) (Sommerhoff, 1975; Winsor
et al., 2015a). From their setting these moraines were correlated with the Fiskebanke moraine system to the north
(Funder et al., 2011), and thought to reflect ice-margin readvance. In lack of absolute dating, they were correlated to the
YD cold event in the Greenland ice cores (Weidick et al., 2004 ("Neria Stade"); Roberts et al., 2009). A YD readvance
for this part of the ice sheet was also suggested by modelling, which indicated that the ice-margin here retreated from



the shelf edge to the present coastline in the Bølling-Allerød period, but then returned to the shelf during YD (Simpson
et al., 2009; Lecavalier et al., 2014). A grounding zone wedge in Fiskenæsset trough also points to a stillstand or
readvance of the glacier front for an unknown period of time during deglaciation (Fig. 1, Ryan et al., 2016). The
significance of these features is discussed below in the light of the new cosmogenic exposure dates.

**3. Field and laboratory methods**

In the field, the samples were collected from knobs in the glacially sculptured terrain. Unfortunately there were hardly
any erratic boulders on the bare rock surfaces, and the samples were generally collected from the bedrock surfaces. This
potentially represents a problem because while boulders were ideally incorporated in the ice in a pristine condition
without previous exposure to cosmic radiation, the glacial erosion of the bedrock surface may not have been deep
enough to remove inherited isotopes from older exposures, which may result in overestimation of the age (Briner et al.,
2006; Corbett et al., 2013; Larsen et al., 2014). To minimize the risk for inheritance, we selected sites in the lowland
where the overlying ice would have been thickest and most erosive, but above the marine limit to avoid the risk of
shielding of the rock surface by the sea. From each site we collected 3-4 samples within a radius of 100 m to be sure
that all samples from each locality had been deglaciated at the same time.
Contrary to inheritance other potential sources of error may give too young ages. This may occur if the
surface for some time has been partially shielded from cosmic radiation by vegetation or snow cover (Gosse and
Phillips, 2001). However, it is inconceivable that the rocky tops were ever vegetated, as soil then would have been
washed into the depressions of the glacial sculpture, which was not to be seen (Fig. 2). Also, long lasting deep snow
cover over the tops is unlikely in the stormy climate at the outer coast. Indeed, we experienced heavy snowfall during
the sampling, with thick snow accumulating in hollows, while the tops were left free of snow (Fig. 2). Topographic
shielding from nearby mountains was checked with a clinometer in the field, but in all cases was non-existent.
The laboratory work comprised sample preparation at the University of Buffalo and measurement of $^{10}$Be-
concentrations at the AMS facility at Aarhus University. The laboratory procedure for the preparation followed the
University at Buffalo's protocol (Briner, 2015). Samples were crushed and sieved to 250–500 μm to obtain grains of
single mineral species, then exposed to a magnetic separator to remove the more magnetic minerals and facilitate the
subsequent froth flotation. In addition to flotation, some samples (X1509, X1513, X1521) had to undergo heavy mineral
separation to obtain sufficient amounts of quartz. Before the next step the samples were examined under a microscope
to see if they had been substantially purified. Finally, the samples were etched by hydrogen chloride (HCl) and a
mixture of hydrofluoric and nitric acid (HF/NHO₃) in order to further isolate pure quartz from remaining minerals.
Quartz purity was then verified by inductively coupled plasma optical emission spectroscopy at the University of
Colorado. At the University at Buffalo, pure quartz samples were fully dissolved with a $^{9}$Be carrier and Be(OH)$_2$ was
isolated through column separation. After ignition, the final BeO was measured by accelerator mass spectrometry
(AMS) in Aarhus
The ages were calculated with the CRONUS-Earth online calculator (Balco et al., 2008), using the $^{10}$Be/$^{9}$Be-
ratio measured by the AMS subtracted the processed blank ratio. The processed blank ratio was $2.10 \times 10^{-15}$ and the
blank-corrected sample ratios ranged from $0.76 \times 10^{-13}$ to $2.58 \times 10^{-13}$. The Arctic production rate defined by Young et



al. (2013) and the time-invariant scaling scheme for spallation processes given by (Lal, 1991) and Stone (2000) were
applied. The time-invariant scaling scheme does not incorporate variations in past geomagnetic field strength, but these
usually only affects younger samples, at c. 10 ka, by 1% (Nishiizumi et al., 2007). The maximum deviation between
different scaling schemes in this material is around 1%, so they generally, provide consistent ages and do not affect the
relative chronology. We used a rock density of 2.65 g cm$^3$ and made no correction for potential surface erosion or
snow/vegetation cover. The study area has undergone glacioisostatic uplift since the deglaciation, and this may
potentially influence the $^{10}$Be ages. However, the uplift history of each sample site is poorly constrained and this effect
is partly counteracted by atmospheric pressure changes in ice-marginal settings (c.f. Young et al. 2020). Accordingly,
we present $^{10}$Be ages without correcting for glacioisostatic uplift, similar to most other $^{10}$Be studies from Greenland.
Individual $^{10}$Be ages are presented with their 1-sigma analytical uncertainties, which include the uncertainty in the blank
correction, the "internal" uncertainty (Table 1). When we compare our $^{10}$Be ages with $^{14}$C ages or climate records we
include the production rate uncertainty, the "external" uncertainty (Balco et al., 2008).
Previously published $^{14}$C ages have been re-calibrated using the Intcal13 calibration programme (Reimer et
al., 2013). Following the procedure adopted for dates on marine shells from Greenland ages on marine shells have been
corrected for a reservoir effect of ΔR = 0 for western and -150 yr for eastern Greenland, based on dating modern pre-
bomb shells (e.g. Mörner and Funder, 1990), acknowledging that significant, but unknown, changes in the reservoir
effect may potentially/likely have affected the ages especially in the turbulent millennia during the early deglaciation
phases (e.g. Andrews et al., 2018).

**4. Results**

As noted above a problem in using bedrock samples is the risk of inheritance in the rock surface. The knobby terrain is
evidence that basal glacial erosion was not uniform, but varies in intensity even over small distances. Therefore the
amount of inheritance may also vary over short distances, resulting in a spread of old ages (Corbett et al., 2013). We
therefore consider a spread of old ages as "inheritance outliers", while the mean of clustered younger ages gives the
most reliable deglaciation age. Where there is no overlap, we regard the youngest age as a minimum age for
deglaciation. At each site our new ages are compared with previously published cosmogenic and $^{14}$C results from the
coastal areas. Much of this information has recently been reviewed by Sinclair et al. (2016). The six sites are described
below, and the results are shown in Table 1 and Fig. 3.
**4.1 Buksefjord**
This site is located on a small island at the outer margin of the strandflat, c. 15 km from the coast and midway between
the mouth of Ameralikfjorden and Buksefjorden. The three bedrock samples from this locality were collected at
altitudes between 102 and 118 m a.s.l. and gave ages, with their internal uncertainties, of 14.4 ± 0.8 ka (X1524), 12.5 ±
0.6 ka (X1525) and 12.1 ± 0.6 ka (X1526). We regard the oldest age as an inheritance outlier, while the two youngest
ages have overlapping internal uncertainty, yielding the age of 12.3 ± 0.2 ka in mid-Younger Dryas times as the best
estimate of the time of deglaciation at this site.



On the coastal mountains c. 10 km to the east, [10]Be dates from between 82 and 360 m a.s.l. gave an average
deglaciation age 10.7 ± 0.6 ka (Larsen et al., 2014). At the mouth of the Nuuk Fjord Complex, 30 km to the north,
marine shells in the coastal area have been dated to 11.4 and 10.7 cal. ka BP (Weidick, 1976; Larsen et al., 2014), while
[10]Be ages close to Nuuk showed deglaciation at c. 11 ka (Winsor et al., 2015b). This is somewhat later than our
deglaciation date in the skaegaard, indicating that the fjord glaciers lingered on in their troughs while the adjacent
coastal areas became ice free (Larsen et al., 2014).

**4.2. Fiskenæsset**
Three bedrock samples were collected in the outer archipelago c. 6 km from the coast, from a small ice-scoured island
c. 15 kilometres west of the Fiskenæsset settlement. The samples were collected at altitudes of 75-76 m a.s.l and gave
congruent ages of 13.0 ±1.0 ka (X1521), 12.9 ± 0.7 ka (X1522), and 13.9±0.8 ka (X1523), and the average, 13.3 ±0.5
ka in late Allerød times, gives a robust estimate of the time of deglaciation at this site. This is the oldest deglaciation
date of our sites, lying well inside the Fiskebanke moraines in this area (Fig. 1).
Further north, at the Sermilik Fjord [14]C dates of marine molluscs show that the initial marine transgression
and retreat of the ice stream from the outer coast here began before 10.5 cal. ka BP, and [10]Be ages from 450 m a.s.l. on
coastal mountains show that the coast became ice free at c. 10.6 ka (Larsen et al. 2014). Even though these ages are
minimum constraints for deglaciation, it is not likely that they postdate the deglaciation of the outer archipelago with
2000 years. This indicates that also here the major outlet glaciers reached the inner shelf, while adjacent areas had been
ice free for some time.

**4.3. Ravns Storø**
Four samples were collected on the island of Ravns Storø, in the middle of the archipelago, c. 5 km from the coast. The
samples were collected at elevations between 189 and 209 m a.s.l. The 4 samples were collected within a radius of 200
m, but the ages show a spread of more than 5000 years: 13.7 ±0.8 ka (X1519), 17.0 ±1.7 ka (X1520), 11.9 ±0.7 ka
(X9364) and 11.5 ±0.8 ka (X9365). The two youngest ages, including our only boulder sample (sample X9365), have
overlapping internal uncertainties, and we consider their average, 11.7±0.2 ka, as the best estimate for the time of
deglaciation at this site, while the oldest ages are inheritance outliers. From this area there is no supporting information
on deglaciation history.

**4.3. Avigaat**
Three samples from the bedrock surface were collected from an islet in the inner archipelago, c. 3 km from the coast
and the abandoned Avigaat settlement. The samples were taken at elevations between 42 and 47 m a.s.l. and gave ages
of 13.7 ± 1.1 ka (X1516), 12.0 ± 0.5 ka (X1517) and 10.3 ± 2.5 ka (X1518). The variation and uncertainties are very
large, but overlap, and we consider the average of 12.0 ± 1.6 as the best estimate for deglaciation at this site. Some
support for this comes from a [14]C age of 11.3 cal. ka BP for basal gyttja in a lake in coastal Nerutussoq fjord to the
south, giving a minimum age for deglaciation at this site (Kelly and Funder, 1974).



### 4.4. Paamiut

This site is located on a small ice-scoured island on the inner archipelago, c. 5 km from the coast and close to the mouth of Kuanersoq fjord and the town of Paamiut. Here three bedrock samples between 60 and 65 m a.s.l - X1513, X1514 and X1515 – are dated to 12.5 ±0.7, 12.1 ±1.3 and 12.0 ±1.4 ka. All three overlap within the internal uncertainty, and the average, 12.2 ± 0.2 ka, indicates that the ice-margin here was retreating over the inner shelf in mid-Younger Dryas times.

Around Paamiut and Kuanersoq several studies have supplied both [10]Be and [14]C deglaciation dates for the outer fjord. As could be expected these deglaciation dates from further inland are somewhat younger than ours. [10]Be dates from Kuanersoq indicate thinning of the ice stream in the fjord since 11.7 ka, and, by extrapolation, retreat from the fjord mouth at c.11.2 ka (Winsor et al., 2015b). From a [14]C age of 11.0 cal. ka BP for basal gyttja in a lake 8 km from our samples and well below the local marine limit Woodroffe et al. (2014) suggested that deglaciation could not have been much earlier than 11 ka. These results from nearby coastal localities therefore indicate deglaciation c. 1000 years later than at our site. Much of this work concerned the ice stream in Kuanersoq, while our samples come from the open coast to the south, and we suggest that an ice stream in the Kuanersoq trough remained at the inner shelf while the adjacent coastal areas became ice free.

### 4.5. Sermiligaarsuk

From a small island in the inner archipelago, c. 2 km from the coast and 12 km south of the Sermiligaarsuk fjord we collected two samples at altitudes from 56 to 61 m a.s.l. One from bedrock and one from a 1 m boulder (X1507). The two samples have widely scattered ages of 10.9 ±2.3 ka (X1507) and 14.7 ±0.8 ka (X1509) with a very large uncertainty especially in the youngest age. The oldest age is unrealistic for deglaciation and interpreted as an inheritance outlier, and the age of 10.9 ±2.3 ka (sample X1507), one of our few boulder dates, is considered as a minimum for deglaciation at this site. This age is the youngest for deglaciation of the inner shelf, but the island is also closer to the coast than any of the other sites.

Marine shells below the marine limit in the nearby outer Sermiligaarsuk Fjord gave [14]C age of 9.7 cal. ka BP, as a minimum for deglaciation at this site (Weidick et al., 2004).

## 5. Discussion

### 5.1. Overview of results from Southwest Greenland

According to the criteria outlined above, two of the sites, Paamiut and Fiskenæsset, apparently show no inheritance, and give trustworthy deglaciation ages in late Allerød and mid YD times. Also Avigaat has overlapping uncertainties, indicating deglaciation in late YD times, but with a large uncertainty. At two sites, Buksefjord and Ravns Storø, one or two samples in each are apparently affected by inheritance, but the remaining cluster and indicate deglaciation during mid-late YD. Finally, at Sermiligarssuk only one sample is considered free of inheritance, yielding a best estimate for deglaciation in the Early Holocene. The results therefore show that deglaciation on the inner shelf in this part of Greenland was underway during the Allerød-YD period, and the ice sheet margin had reached a position close to the present coastline, 60-40 km from the shelf break. The results also show that even in the archipelago, which is a result of



intense erosion by warm based ice, inheritance from previous exposure is a problem, as seen from the six inheritance
outliers. At three localities, Paamiut, Fiskenæsset and Buksefjord, our ages are considerably older – up to 2 ka – than
deglaciation ages obtained at nearby fjord mouths. This suggests that while the coastal areas became ice free, ice
streams remained in the major drainage troughs and reached the inner shelf until the Early Holocene, as also shown
previously for ice streams in Disko Bugt and the Sisimiut area to the north (Roberts et al., 2009; Jennings et al., 2014).
Together the results show that the ice sheet margin in this area was retreating on the inner shelf during YD,
and probably was close to the coast already in late Allerød times. We find no evidence for YD stillstand/readvance, as
has previously been suggested from this area (Weidick et al., 2004). The Fiskebanke moraines further out on the shelf
as well as the grounding zone wedge in Fiskenæsset trough may reflect long-lasting stillstand/readvance, but they are
apparently older than YD (Fig. 1).

## 5.2. YD ice-margins in Greenland

Although it may seem surprising that the ice-margin in this area apparently retreated during the cold YD oscillation this
seems not to be unique to our area. From a recent review of YD ice-margins in Greenland Larsen et al. (2016)
concluded that ice-margin retreat indeed characterised most areas with a dated record going back through or into YD.
This is well constrained by $^{10}$Be dating in coastal areas and $^{14}$C dated marine cores in and outside major transect
troughs, and applies to areas in western, eastern and southernmost Greenland (Fig 4). More recently this has been
corroborated by new evidence from the Disko Bugt shelf (Hogan et al., 2016; Oksman et al., 2017) , and from east
Greenland (Levy et al., 2016; Andrews et al., 2018; Dyke et al., 2018; Rainsley et al., 2018). However, this has recently
been contested by evidence from multibeam bathymetry in major transverse troughs on the shelf, notably the occurrence
of Grounding Zone Wedges (GZWs) as discussed below.
In the dated records the most dramatic retreat occurred on the shelf at Disko Bugt, where the ice stream
apparently retreated over more than 200 km from an Allerød-Early YD position at near the shelf break (Fig. 4), but the
retreat was punctuated by periods of topographically conditioned stillstand and a spectacular, but brief, non-climatic
readvance (e.g. O'Cofaigh et al., 2013; Hogan et al., 2016). A similar history is recorded on the east side of Greenland
(Fig. 4), in Kangerlussuaq trough and fjord, where not only the shelf, but also the better part of the fjord became ice free
during YD (e.g. Andrews et al., 2018).
In these areas, as well as in southeast Greenland and in our study area the retreat occurred on the shelf (Dyke
et al., 2018). In other areas, northernmost and southernmost Greenland and the Scoresby Sund fjord complex, the ice-
margin had already retreated behind the present coastline before YD (e.g. Björck et al., 2002; Larsen et al., 2016; Levy
et al., 2016). These records therefore generally show ice-margin retreat during YD, but with large differences in rate
and areal extent from area to area. None of these records show evidence of ice-margin readvance during the initial YD
cooling.

## 5.3 YD moraines on the shelf?

A YD readvance of the ice-margin has previously been suggested, notably from western Greenland. Here, since their
first discovery, the Hellefisk and Fiskebanke moraines on the west Greenland shelf have played a prominent role in the
discussion of early deglaciation history (Fig.4) (e.g. Kelly, 1985; Funder et al., 2011; Hogan et al., 2016). The



outermost and oldest, the Hellefisk moraine system, runs along the shelf break for 200 km at a depth of c. 200 m, c. 120
km from the coast and consists of swarms of up to 100 m high ridges (Brett and Zarudski, 1979). To the east of this,
halfway towards the coast, the younger Fiskebanke moraines mentioned above impinge on the inner side of the fishing
banks c. 40 km from the coast. This deglaciation-stage is composed of single ridges, which occur on and off, on the
inner banks and along the sides of transverse troughs for a distance of c. 500 km along the coast (Fig. 4). Although
undated, the two moraine systems have generally been regarded as climate signals for two distinct periods of cooling
with the Hellefisk moraines marking LGM and the Fiskebanke moraines possibly a YD readvance, seen also in models
(Weidick et al., 2004; Roberts et al., 2009; Simpson et al., 2009; Lecavalier et al., 2014).

Recently Hogan et al. (2016), from [14]C dates in marine cores, found that in their study area south of Disko

Bugt, the oldest moraines, the Hellefisk moraine, dated, not from LGM, but marked the establishment of a deglacial
calving bay during late YD, while, as noted above, in our study area to the south, the "younger" Fiskebanke moraine is
older than YD. Although still sparse, these new results suggest that the two moraine systems may not be climatic signals
of distinct periods of cooling, but rather that they are metachronous expressions of differences in ice-margin dynamics
during deglaciation. The Hellefisk moraines, formed by an ice-margin at the shelf break and exposed to the ocean,
which was especially sensitive to changes in ocean currents, sea ice, and sea level, all of which may vary locally with
the depth of the grounding line and extent of exposure to the ocean as described by Jennings et al. (2017). Therefore the
timing of retreat from the shelf break varies with several millennia from trough to trough (Winsor et al., 2015a;
Jennings et al., 2017), determined by local conditions as much as climate change. The Fiskebanke moraines, on the
other hand, were deposited by an ice-margin entrenched in the coastal trough with limited exposure to the ocean, and
more resilient to these agents and able to withstand a warming climate. These moraines therefore may reflect, not a
response to cooling, but the time at which the retreating ice-margin reached the inner trough and became stabilised for a
longer period, as discussed for Melville Bugt, Northwest Greenland, by Newton et al. (2017).

Only in northernmost Greenland did glaciers from a local ice cap advance/retreat during the YD

cooling/warming, as documented by a combination of cosmogenic [10]Be, [14]C and OSL (Optically Stimulated
Luminescence) dating (Larsen et al., 2016).

The dated records from the GrIS margin therefore generally show YD ice-margin retreat or thinning in all

parts of Greenland, but both speed and extent varied greatly from area to area and trough to trough. Clear evidence for
readvance/stillstand related to the YD cold oscillation is missing in these records, and moraines on the shelf may not
register climate change, but glacio-dynamic response to local topography and bathymetry.

### 5.4. Grounding Zone Wedges and YD readvance/stillstand?

Evidence for readvance or long-lasting stillstand has recently been suggested from several major transect troughs. This
is based on high resolution multibeam bathymetry, revealing a large variety of glacial landforms in transect troughs
formed during retreat of major ice streams. Notably, the occurrence of large GZWs has been suggested to reflect long
lasting stillstand on mid shelf. GZWs are wedge-shaped sediment accumulations deposited at the front of an ice stream
during a period of stability (e.g. Dowdeswell and Fugelli, 2012). They have now been observed in most of the
investigated troughs, and, although not dated, prominent GZWs have tentatively been referred to YD, based on the
assumption that they correlate with cold periods in the ice core temperature record (Sheldon et al., 2016; Slabon et al.,





2016; Arndt et al., 2017; Newton et al., 2017; Arndt, 2018). However, in areas with both a [10]Be - [14]C dated and a
"climate-correlated" GZW record for YD, there is a striking difference between the two dating-approaches.

In the Uummannaq trough deglaciation of the shelf began before 15 ka and by c. 11.5 cal. ka BP the large ice

stream in the trough had disintegrated into fjord glaciers with their front close to the present ice-margin (e.g. Jennings et
al., 2017). However, there are two very different views on what happened in the intervening 3.5 millennia. Based on
exposure ages on coastal mountains and [14]C dates in the fjords Roberts et al. (2013) found that the large Uummannaq
ice stream had retreated from the trough and into the fjords during YD, controlled by topography and bathymetry (Fig.
4). In contrast to this, Sheldon et al. (2016), from a series of marine cores and a prominent GZW in the transect trough,
suggested that the ice stream was stabilised for 2 ka on the outer shelf, 150 km further away from the coast (Fig. 4), and
remained here through Allerød and YD and into the Preboreal. This was based on extrapolation from a [14]C age and
correlation with the ice core temperature record.

An even more striking discrepancy between the two dating approaches is in the Scoresby Sund fjord

complex. Here Greenland's most spectacular moraines show that already in late Allerød times the outlet ice streams had
retreated c. 100 km into the fjord system, and during YD and into Preboreal times the oscillating ice-margin retreated in
mid fjord, until rapid retreat in the narrow inner fjords began in Preboreal times. This is shown by Greenland's highest
concentration of [10]Be and [14]C dates from this period (Denton et al., 2005; Kelly et al., 2008; Hall et al., 2010; Levy et
al., 2016). However, from high resolution bathymetry Arndt (2018) advocated a very different sequence of events. In
this model, lineaments in the wide Hall Bredning were interpreted to show that fast flowing ice during YD advanced
and formed the very large Kap Brewster moraine at the mouth of Scoresby Sund, 100 km outside the Allerød-YD
moraines (Funder et al., 1998). The new YD age for this event is based on correlation with the ice core temperature
record and the misconception that the mid fjord moraines were formed by local mountain glaciers with no relevance for
GrIS. However, these significant moraines were without doubt formed both by local glaciers and by several major
outlets from the GrIS (e.g. Denton et al., 2005; Levy et al., 2016), ruling out any possibility that ice streams should have
advanced 100 km beyond the moraines in YD. The advance of fast flowing ice must therefore be older, leaving the
interesting question: when?

Similar outbursts of fast flowing ice streams, reaching mid shelf GZWs were recorded also further north in

transect troughs at Kong Oscar Fjord, Kejser Franz Joseph Fjord and the wide shelf of Northeast Greenland, and dated
under the same token to YD (Arndt et al., 2017; Arndt, 2018). Also here it has been overlooked that mid fjord moraines
in these fjords, 100 km behind the mid shelf GZWs, previously have been dated to late YD/earliest Preboreal (Hjort,
1979). Reconciling these two datasets would imply an extraordinarily dynamic behaviour of the ice-margin along the
East Greenland seaboard, with both advances and retreats of more than 100 km within YD in a period with increased
sea ice along the coast (e,g, Fluckiger et al., 2008; Buizert et al., 2018). Constraining the collapse of the putative fast
flowing ice streams in all troughs, the authors have chosen to use uncalibrated [14]C ages from land, delaying the
deglaciation c. 1.5 ka from late YD to the end of the Preboreal relative to ice core chronology.

In Northwest Greenland mid shelf GZWs with a length of more than 100 km have been recorded in transect

troughs on the shelf in Melville Bugt (Slabon et al., 2016; Newton et al., 2017). From analogy with the "climate-
correlated" Uummannaq GZW they were tentatively referred to YD, although non-climatic, bathymetric conditions may
also have determined their position (Newton et al., 2017).



The $^{14}$C and $^{10}$Be dated and the "climate-correlated" records therefore paint two very different pictures of ice-
margin behaviour in Greenland during YD. Not surprisingly the climate-correlated records of GZWs and submarine
landforms show high correlation of ice-margin behaviour with ice core temperatures. More surprisingly, none of the $^{14}$C
and $^{10}$Be dated records from marine sediment cores and cosmogenic deglaciation dates on land show a clear signal
neither of initial YD cooling, nor of the abrupt terminal warming, but mostly point to retreat of the ice-margin through
YD (Fig. 4). This highlights the need for climate-independent dating of the submarine landforms and GZWs, to exploit
this rich source of information, and get a better understanding of the ice sheet/climate relation.

### 5.5. Ice-margin retreat during the YD cold oscillation?

To explain the mismatch between YD cooling, and apparent ice-margin retreat, two agents have especially been called
on: advection of warm oceanic subsurface water to the ice-margin, and increased climatic seasonality.
In both cases the sequence of events begins with increased production of meltwater around the North Atlantic
during the Allerød warm period. The fresher and lighter water eventually sealed off the Atlantic surface circulation
from the atmosphere and impeded AMOC. However, warm water from the subtropical areas was still driven into the
North Atlantic, but now as subsurface currents (Marcott et al., 2011; Ezat et al., 2014). The subsurface water followed
the path of the present North Atlantic surface circulation in the Irminger Current running south along Southeast
Greenland, continuing around Greenland's southern tip, and heading northwards as the West Greenland Current (Fig.
1). Along the Greenland shelf the warm Atlantic subsurface water was present and caused ice-margin retreat in
Southeast and West Greenland at 15-16 cal. ka BP, and it was continuously present at the southeast Greenland shelf
edge through Bølling-Allerød and YD times (Kuijpers et al., 2003; Knutz et al., 2011; Jennings et al., 2017; Andrews et
al., 2018).
Today warm subsurface water from these currents, below a cap of fresher water, causes extensive melting of
floating outlet glaciers in Greenland (e.g. Mayer et al., 2000; Motyka et al., 2011), and during the early phase of
deglaciation when the GrIS had its entire margin on the shelf it was especially sensitive to the advection of warm
subsurface water, causing ice-margin retreat even when temperatures were dropping as discussed extensively in the
literature (Kuijpers et al., 2003; Jennings et al., 2006; Knutz et al., 2011; Rinterknecht et al., 2014; Winsor et al., 2015b;
Sheldon et al., 2016; Sinclair et al., 2016; Jennings et al., 2017; Oksman et al., 2017; Andrews et al., 2018; Dyke et al.,
2018; Rainsley et al., 2018)..
Crucial to the impact of the warm water is the depth of the grounding line at the ice-margin, and the
accessibility for the warm subsurface water. This is again dependent on local bathymetry and – in the troughs – of the
type of connection to the open ocean, which in each area may control the impact of the warm water on the ice-margin,
as well as the impact from changing sea level and presence or absence of buttressing sea ice. This may explain why the
deglaciation of the shelf and troughs had such a different character even between neighbouring troughs, such as
between the rapid deglaciation of the Kangerlussuaq trough and the much slower deglaciation along the adjacent coast
to the south (Dyke et al., 2018). It may also explain the differences in the timing of onset of deglaciation in the
neighbouring Nuussuaq and Disko troughs in West Greenland as discussed by Jennings et al. (2017).
Also increased seasonality owes to the meltwater cap over the North Atlantic, and reduced AMOC. The
fresher meltwater-diluted water seals off the ocean surface and cuts off the ocean-atmosphere heat exchange, especially



in winter (Denton et al., 2005; Hall et al., 2008; 2010; Buizert et al., 2014; Levy et al., 2016; Buizert et al., 2018). This
results in very cold and arid winters and increase in extent and duration of sea ice, while summer temperatures – which
primarily determines a glacier's mass balance - are less affected and may even warm up (Björck et al., 2002).
The role of seasonality has recently been investigated in a model where the deglacial ice-core temperature
records in three ice cores are combined with simulated seasonal air temperatures for the whole of Greenland, enabling
assessment of variations in seasonality in time and space (Buizert et al., 2018). The varied seasonality model is largely
independent of field observations, and it is therefore interesting to note that on some points where it deviates from the
previous Huy models (e.g. Lecavalier et al., 2014), do seem be supported by the field observations: 1) The varied
seasonality model indicates that the deglacial rate of mass loss peaked already in the Bølling-Allerød period, and not in
the early Holocene. As shown above, recent results from many parts of Greenland show that large areas of the shelf had
been cleared of ice already before YD; 2). Both model-approaches show mass loss of ice from the GrIS throughout YD
although as noted above the Huy models suggest YD readvance on the shelf in West Greenland. However, the dramatic
temperature changes in the ice core records at the beginning and end of YD, are muted in the varied seasonality model,
in agreement with the field evidence where these rapid changes do not seem to have left a clear signal in the ice-margin
behaviour; 3) In the varied seasonality model the amplitude of YD seasonality decreases from southeast to northwest,
away from the source of AMOC, as underlined by the YD record from northernmost Greenland – the area farthest away
from North Atlantic ocean currents.
In summary, the apparent contradiction between ice core temperature records, where temperatures dropped
dramatically at the onset of YD, and the dated glacial record where glaciers in most parts retreated, may be explained by
the effect of warm subsurface water on the ice-margin, which was, all around Greenland, located on the shelf during the
early phase of deglaciation, while local topographic and bathymetric conditions controlled the access of warm water to
the ice-margin. Increased seasonality owes to increase in distribution and duration of winter sea ice, and the YD
temperature drop in the ice cores was due to a large extent to lowering of winter temperatures with little impact on the
ice-margin, but a large effect on distribution and duration of sea ice. This may also explain why neither the rapid initial
YD cooling nor the abrupt warming at the end left a clear track in the dated records.

## 6.  Conclusions

[10]Be dates from the inner shelf over a stretch of 300 km along the open coast of Southwest Greenland indicate that the
ice-margin here was retreating and close to the coast through YD. Mid-shelf moraines and a GZW further out on the
shelf may reflect earlier stillstand/readvance of the ice-margin.
A survey of [10]Be and [14]C dated records, going back through YD elsewhere in Greenland - south, east and
west - shows that also here the ice-margin was retreating, but with large differences in speed and extent even between
neighbouring basins. Only in northernmost Greenland did glaciers from a local ice cap apparently advance/retreat at the
beginning/end of YD. The survey also shows that the shelf in areas both in north, south, east and west Greenland had
essentially been cleared of ice before YD, giving some support to recent modelling, implying that the main loss of ice
since LGM in the GrIS was in the Bølling-Allerød period, not in the Holocene.
The apparent mismatch between the ice core temperature record and the ice-margin behaviour in these
records, may be explained by the circumstance that during LGM the GrIS, contrary to other large ice sheets, around the



whole perimeter was standing on the shelf and especially sensitive to changes in ocean currents, sea level, and sea ice
distribution and thickness. From this, the large variability in ice-margin behaviour between areas may be explained by
local differences in topography, bathymetry and basin geometry, allowing or hampering access of the ocean to the ice-
margin.

Recently, high resolution bathymetry has supplied a wealth of data on ice stream dynamics during
deglaciation. However, this evidence is essentially dated only by "climate-correlation". To tap this rich source of
information and get a better understanding of the ice sheet/climate relation, climate-independent dating of the
submarine features is badly needed.

To conclude, we fully subscribe to the contention by Andrews et al. (2018, p. 16): "The use of the GrIS's
isotopic records as a one-to-one template for coeval changes in glacier and ocean response potentially ignores the
different response timescales between the atmosphere, oceans and cryosphere" – with a bearing also on the future.

**Author contributions**. SF, KK and AB conceptualised the project and carried out the work in the field. AS did the
laboratory work, critical assessment and compilation of data under supervision of and according to methodologies
developed by JB, NL, AS and JO. Visualization owes to NL and AB. The writing and editing was made by SF in close
cooperation with NL and AB. KK was responsible for the funding acquisition.

**Acknowledgement**
This study was made possible by grants from Danish Council of Natural Sciences (FNU) and Danish National Science
Foundation to carry-out field and laboratory work. It would not have been possible without generous support from the
Danish Navy and the ship "HDMS Knud Rasmussen" with its skipper and his extremely help- and joyful crew.




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




Table 1. Summary of [10]Be data from Southwest Greenland.

| Sample ID | Latitude (N) | Longitude (W) | Elevation (m a.s.l.) | Sample type† | Shielding factor | Thickness (cm) | Quartz (g) | [9]Be carrier weight (g) | [10]Be conc. (atoms/ g)×10⁴ | [10]Be uncert. (atoms/ g)×10⁴ | [10]Be age (ka) internal (external) uncertainties†† |
|---|---|---|---|---|---|---|---|---|---|---|---|
| **Buksefjord** | | | | | | | | | | | |
| X1524 | 63.83957 | 51.73826 | 118 | bedrock | 1 | 4.5 | 40.45 | 0.6067 | 6.88 | 0.36 | 14.43 ± 0.76 (1.03) |
| X1525 | 63.83970 | 51.73851 | 117 | bedrock | 1 | 5.5 | 33.19 | 0.6082 | 5.90 | 0.30 | *12.48 ± 0.64 (0.88)* |
| X1526 | 63.83967 | 51.73839 | 102 | bedrock | 1 | 6 | 40.13 | 0.6086 | 5.59 | 0.26 | *12.06 ± 0.57 (0.82)* |
| | | | Calculated average (number of samples out of total) | | | | | | | | *12.3 ±0.4 (n=2/3)* |
| **Fiskenæsset** | | | | | | | | | | | |
| X1521 | 63.04961 | 50.99505 | 76 | bedrock | 0.999962 | 4.5 | 21.16 | 0.6068 | 5.93 | 0.43 | *12.98 ± 0.95 (1.14)* |
| X1522 | 63.05008 | 50.99449 | 75 | bedrock | 0.999969 | 5.5 | 26.75 | 0.6074 | 5.85 | 0.31 | *12.93 ± 0.68 (0.93)* |
| X1523 | 63.05016 | 50.99454 | 76 | bedrock | 0.999969 | 5.5 | 36.35 | 0.6049 | 6.31 | 0.36 | *13.92 ± 0.81 (1.05)* |
| | | | Calculated average (number of samples out of total) | | | | | | | | *13.3 ±0.6 (n=3/3)* |
| **Ravns Storø** | | | | | | | | | | | |
| X1519 | 62.71573 | 50.40947 | 193 | bedrock | 0.999986 | 7 | 35.09 | 0.6074 | 6.95 | 0.38 | 13.71 ± 0.76 (1.01) |
| X1520 | 62.71573 | 50.40947 | 189 | bedrock | 0.999986 | 6 | 45.21 | 0.6083 | 8.66 | 0.87 | 17.03 ± 1.72 (1.91) |
| X9364 | 62.71799 | 50.41719 | 209 | bedrock | 1 | 4.5 | 34.47 | 0.6092 | 6.26 | 0.37 | *11.91 ± 0.70 (0.91)* |
| X9365 | 62.71770 | 50.41629 | 208 | boulder | 1 | 4.5 | 39.87 | 0.613 | 6.05 | 0.43 | *11.52 ± 0.81 (0.99)* |
| | | | Calculated average (number of samples out of total) | | | | | | | | *11.7 ±0.2 (n=2/4)* |
| **Avigaat** | | | | | | | | | | | |
| X1516 | 62.17882 | 49.80153 | 47 | bedrock | 1 | 7 | 45.06 | 0.6062 | 5.94 | 0.48 | *13.68 ± 1.11 (1.29)* |
| X1517 | 62.17888 | 49.80107 | 44 | bedrock | 1 | 6 | 45.08 | 0.6089 | 5.23 | 0.24 | *11.98 ± 0.56 (0.81)* |
| X1518 | 62.17894 | 49.80064 | 42 | bedrock | 1 | 4.5 | 45.26 | 0.608 | 4.55 | 1.10 | *10.31 ± 2.49 (2.54)* |
| | | | Calculated average (number of samples out of total) | | | | | | | | *12.0 ±1.7 (n=3/3)* |
| **Paamiut** | | | | | | | | | | | |
| X1513 | 61.85744 | 49.53121 | 65 | bedrock | 1 | 6 | 32.31 | 0.6111 | 5.57 | 0.29 | *12.45 ± 0.66 (0.89)* |
| X1514 | 61.85734 | 49.53098 | 61 | bedrock | 1 | 6.5 | 25.48 | 0.6086 | 5.34 | 0.56 | *12.05 ± 1.28 (1.40)* |
| X1515 | 61.85708 | 49.53045 | 60 | bedrock | 1 | 5.5 | 35.44 | 0.607 | 5.36 | 0.62 | *12.01 ± 1.40 (1.52)* |
| | | | Calculated average (number of samples out of total) | | | | | | | | *12.2 ±0.2 (n=3/3)* |
| **Sermiligarsuk** | | | | | | | | | | | |
| X1507 | 61.32122 | 48.86104 | 57 | boulder | 0.999672 | 6 | 33.35 | 0.6086 | 4.81 | 1.02 | 10.86 ± 2.32 (2.38) |
| X1509 | 61.32136 | 48.86013 | 61 | bedrock | 0.999704 | 5.5 | 24.76 | 0.5672 | 6.55 | 0.37 | *14.66 ± 0.84 (1.10)* |
| | | | Best estimate for age of deglaciation | | | | | | | | *10.9 ±2.3 (n=1/1)* |

†: All samples are coarse grained orthogneiss
†† : Italics: used in average/best estimate (see text)

[10]Be ages were calculated using the online exposure age calculator formerly known as the CRONUS-Earthonline exposure calculator v.3 (Balco et al., 2008), the Baffin Island production rate of 3.96 ± 0.07 at g-1 a-1 (regional SLHL) (Young et al., 2013), and the St scaling scheme (Lal, 1991; Stone, 2000) under A rock density of 2.65 g cm⁻³ was used and we assumed zero erosion. Samples were measured using the Beryllium standard 07KNSTD (Nishiizumi et al., 2007).




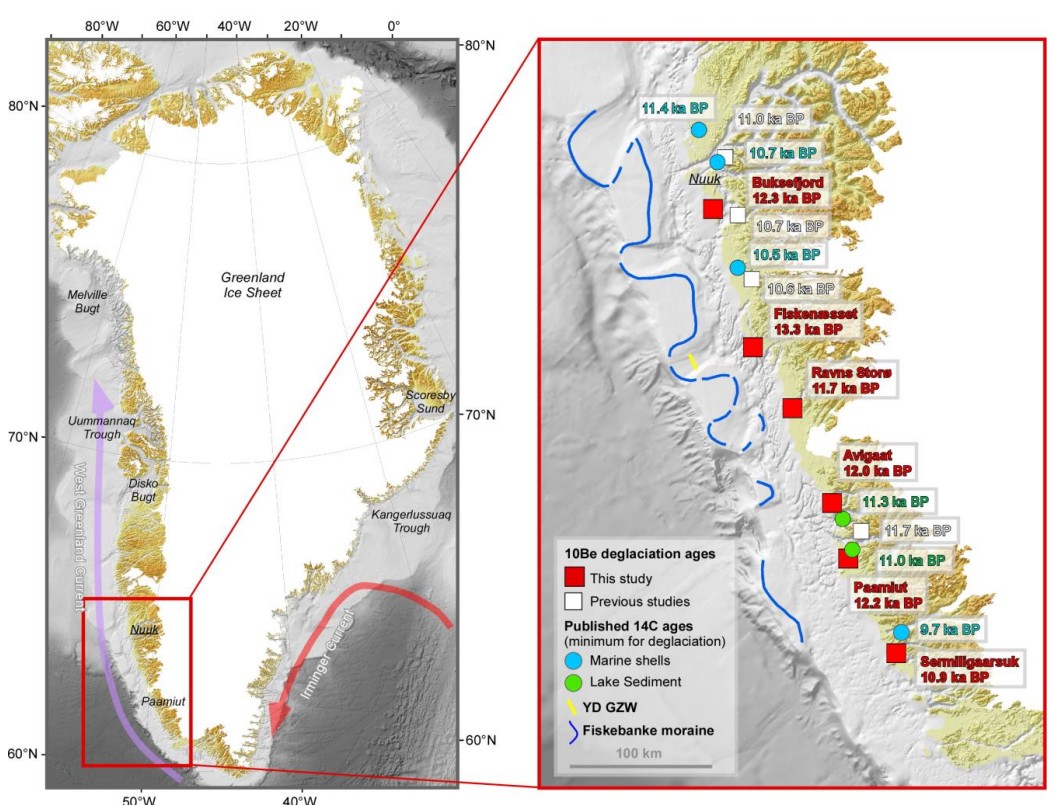

**Fgure 1:** Location of study area and cosmogenic ages from Southwest Greenland, Fiskebanke moraines and selected
published dates relevant to the deglaciation history of coastal Southwest Greenland. For references to previous results
see text. Background map of Greenland and surrounding seas from BedMachine Greenland v.3 (Morlighem et al. 2017.





**Figure 2:** Sampling localities: (a) Buksefjord (Sample X1526, 12.0 ka), (b) Fiskenæsset (Sample X1521, 13.0 ka), (c) Ravns Storø (Sample X1520, 17.0 ka, inheritance), (d) Avigaat (Sample X1518, 10.3 ka), (e) Pamiut (Sample X1515, 12.0 ka), (f) Sermiligarssuk (Sample X1507, 10.9 ka).





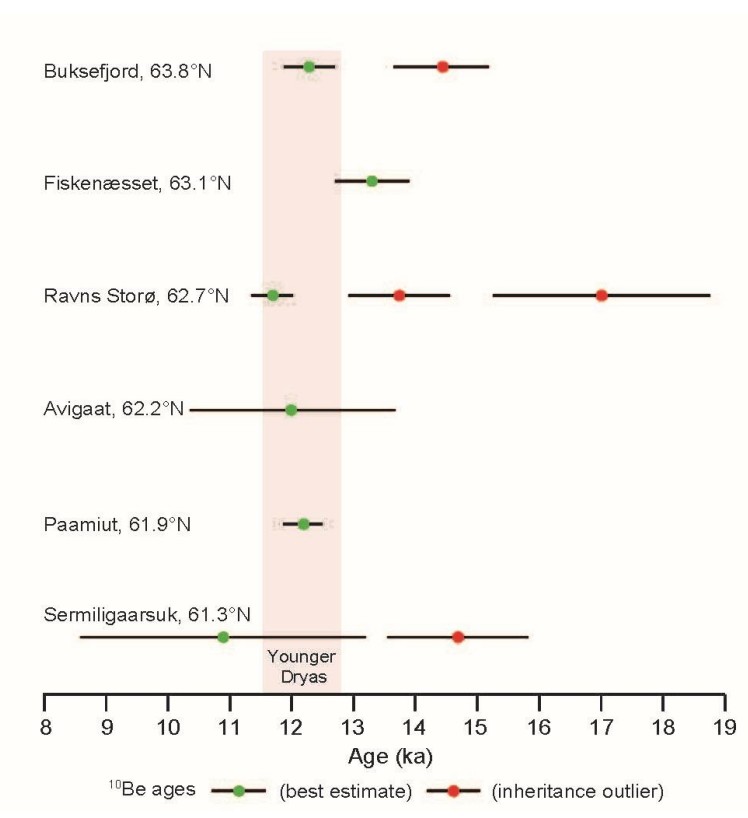

627

**Figure 3:** Cosmogenic ages from Southwest Greenland with external uncertainty.

629

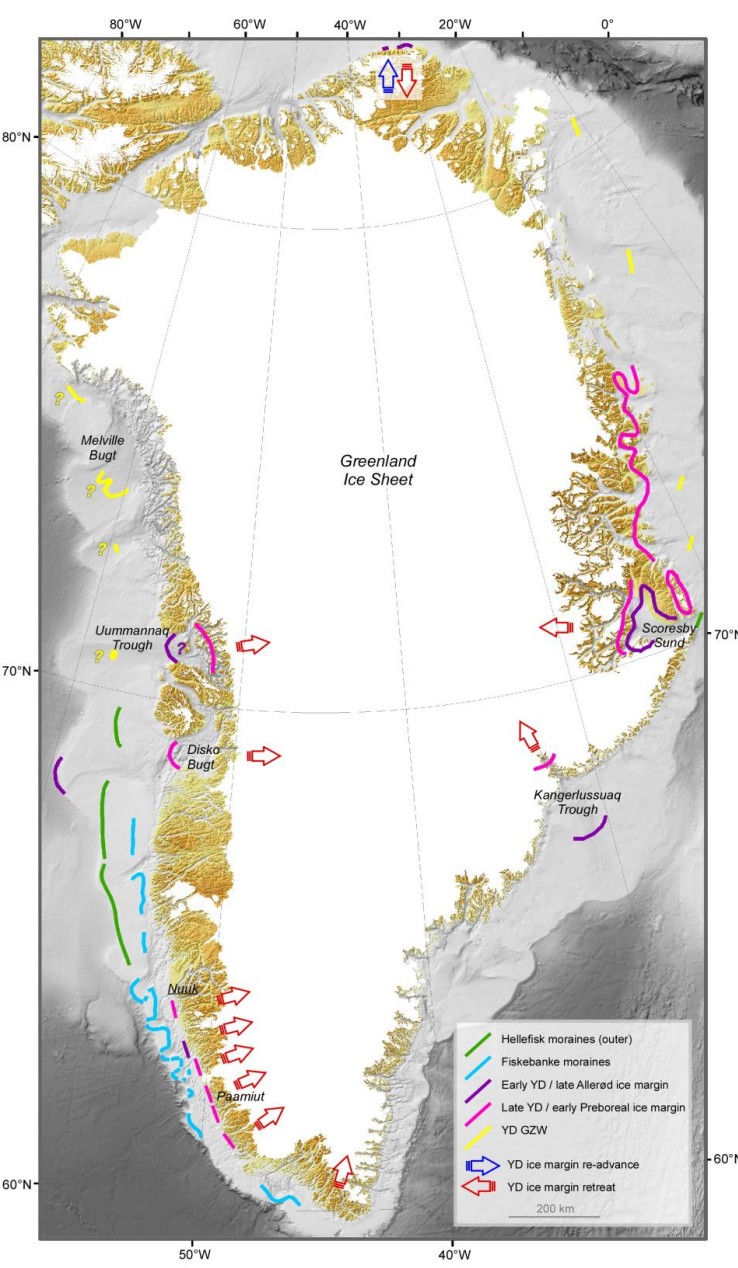

**Figure 4: Deglacial ice-margin** features in Greenland discussed in the text. (The question marks at GZWs apply to age and come from the original literature). Background map of Greenland and surrounding seas from BedMachine Greenland v.3 (Morlighem et al. 2017)