# Peer review of "Younger Dryas ice-margin retreat in Greenland, new evidence"

_Climate of the Past, 2020_

## Short Comment (SC1) · 17 May 2020

This is a fascinating paper, and the authors are to be congratulated. The possibility is important that the new data document oceanic effect of warm subsurface waters on grounding zones during the Younger Dryas. Please note, though, that retreat during the middle and late Younger Dryas (which the bulk of the ages from this paper support) is also consistent with control by surface temperature as recorded in ice cores. Many proxy records of temperature in Greenland, including the deep ice cores, show a relatively large, rapid cooling at the start of the Younger Dryas, then gradual warming (with some oscillations) through the Younger Dryas, and a sharp rise at the end. The simplest translation of that record into ice-marginal position assuming rapid response to temperature yields net retreat during the Younger Dryas (the modeling by

Vacco et al., 2009, doi:10.1016/j.quascirev.20 08.04.018 provides examples of how this behavior can occur). Young et al. (2019, 10.1130/G46058.1 ) found moraines from land-terminating glaciers dating from within the Younger Dryas in the Ahklun Mountains of southern Alaska, and citations therein point to similar behavior elsewhere. It is thus likely that additional data or models will be required to separate the influence of air temperature from ocean temperature.

---

## Short Comment (SC2) · 19 May 2020

This manuscript by Funder et al presents a series of 10Be ages from coastal locations in southwest Greenland that are then used to infer ice-sheet history during the Younger Dryas. This is very much a chronology paper, and many of their arguments are heavily dependent on how one interprets the new 10Be ages. If this entire SW Greenland coastal chronology were based on 10Be ages from erratic boulders perched on bedrock, the author's interpretation of the dataset would be much easier to digest; however, this chronology is almost exclusively from bedrock which presents some challenges. The authors state that these sampling regions were largely devoid of suitable erratic boulders. Thats fine, you can't sample what is not there. But, what to make of standalone 10Be ages from bedrock surfaces can be tricky and I am not sure the

authors here have really given this enough thought.

The primary issue here is that the odds of encountering isotopic inheritance (producing ages that are too old) are higher in bedrock than in erratics, and the authors even acknowledge this. Often times inheritance presents itself as obvious outliers, but there are certainly cases where inheritance is much more subtle and even exists as 'uniform' amounts of inheritance. Again, I think the authors somewhat acknowledge this because they go through great lengths in section 4 to convince the reader that their preferred site by site coastal deglaciation ages are NOT influenced by inheritance even though in some cases their preferred ages of coastal deglaciation are significantly older than the ages of adjacent regions.

Why does this matter? If you take their 10Be ages at face value, then it allows the possibility of the ice margin retreating onto land prior to the start of the Younger Dryas, which in turn the authors use to argue that a major Younger Dryas related oscillation did not occur out on the shelf since everything on the shelf would have to be pre-YD in age. However, a lot of this argument is resting on their 13.3 ka and 12.3 ka sites, Fiskenæsset and Bukesefjord. These ages are noticeably older than the constraints on deglaciation from adjacent areas. The authors take these ages at face value, but I think it is more likely that these suspiciously old sites have slight amount amounts of isotopic inheritance. If the 'true' age of deglaciation is in fact younger and more similar to the remainder of the deglaciation constraints from SW Greenland, the the ice margin is still somewhere out on the shelf during the Younger Dryas. Therefore you cannot rule out a major early or mid-YD oscillation of the ice margin.

Considering all of the 10Be ages from Larsen 2014 and Winsor 2015, and all of the 14C ages from the region, the bulk of the deglaciation chronology around Nuuk (14C + 10Be) suggests deglaciation occurred 10.5 -10.7 ka, with a single older 14C age of around 11.4 ka. With that being the case, I think it is much harder to sell that the Fiskenæsset and Bukesefjord regions deglaciated several thousand years earlier. Note that the region between Fiskenæsset and Bukesefjord also deglaciated at the same

time as the Nuuk region. Considering all of this evidence, coupled with bedrock only sampling, leads me to think that these 10Be ages are influenced by a bit of inheritance.

To be sure, 10Be ages from bedrock can work, but it is typically only in select environments. The most obvious from the region that come to mind are Briner et al (2009; Nature Geoscience) from Baffin Island and Young et al (2013; Quat. Sci. Rev) from Disko Bugt. In both of these cases the 10Be chronologies rely pretty heavily on bedrock samples. Also in both cases the 10Be ages agree with the 14C constraints that exist. In particular, up in Disko Bugt the 14C and 10Be constraints are near identical. This consistency suggests that the bedrock-based 10Be ages are not influenced by inheritance. But what these sites have in common is that the bedrock is located is some of the most erosive environments on Earth where the chances of encountering inheritance should be minimal. On Baffin, samples are from bedrock knobs either right in or adjacent to the Fiord trough (Sam Ford Fiord), and in Disko, these sites are directly adjacent to arguably the fastest glacier on Earth. In this manuscript, the authors are presenting 10Be ages from bedrock in coastal locations *between* former fast-flowing outlets. I wouldn't exactly consider this a recipe for success if you want to avoid inheritance in bedrock surfaces. Even sampling in highly erosive environments does not ensure inheritance-free samples. For example, Hughes et al (2012; Geology) sampled bedrock and erratics in Sermilik Fjord (Helheim) and present pretty convincing evidence that the bedrock samples are influenced by a small (only a few ka) and relatively uniform amount of inheritance. Co-author Briner himself demonstrated the same thing in Norway (2016; Geophysical Research Letters) where on the surface his 10Be ages from erratics are fairly uniform and likely come from what should be an erosive environment. But after considering other constraints, they concluded that their samples are likely influenced by a uniform baseline amount of inheritance equating to only a few ka. As we develop more and more of these 10be-based chronologies, its becoming clear that that small amounts of uniform inheritance exist and bedrock-based ages need to be treated with extra caution.

I look at this dataset and the pre-existing chronology and highly suspect that many of the new 10Be ages in this manuscript have slight isotopic inheritance. And, I think the authors have significantly overreached in their interpretations on what the ice sheet may or may not have done during the YD based on their new 10Be ages. That is not to say these 10Be ages shouldn't be published; they are a valuable contribution that others will certainly build off. It is even possible that these ages are 'correct' but you would need to back the bedrock-only results with matching ages from erratics or any other independent constraint before you launch into sweeping interpretations about the ice margin during YD times.

A few other minor things caught my eye as I read this:

1) It says you used version 3 of the CRONUS calculator for 10Be ages, and a pro-duction rate of 3.96. Version 3 uses an updated treatment of muon production, so the production rate needs to be recast with the same treatment of muon-based produc-tion. So if you used the online calculator, you would have to input all the Baffin Bay calibration data to make your own calibration, and then calculate the ages (there is an option to do this). In this case, the calculator will do the re-casting for you and the production rate is 4.04+-0.07; 3.96 is the number for the old scheme. Your ages should not change, but to avoid confusion, you should list the right PR. Or, you actually used version 2 of the CRONUS code, in which case that 3.96 number still applies (see Young et al., 2020; QSR)

2) The authors spend a lot of time going around Greenland compiling constraints re-lated to the YD ice margin position. Thus I was a bit surprised not to see a fairly robust constraint from near Sisimiut. Young et al (2020; QSR) present several 10Be ages from erratics directly at the coast and from moraine boulders also at the coast, that all date to ~11.6 ka. Therefore the ice margin was out on the shelf during the entirety of the YD.

3) The authors also mention that they do not correct for uplift on 10Be ages because

those effects are offset to some degree by atmospheric pressure variations, and they cite Young et al (2020; QSR). While this is true, a more robust thing you can say here is that the 10Be production rate calibration dataset likely underwent a similar amount of uplift to your known sites and therefore no correction is needed, the correction is essentially 'built in" in this case.

---

## Referee Comment (RC1) · David Ullman (Referee) · 20 Jun 2020

General Comments

This manuscript presents a new cosmogenic 10Be chronology for Greenland Ice Sheet (GrIS) retreat from an archipelago of outer islands off the southwest coast of Greenland. Surface exposure dates from these locations help to constrain the timing of retreat off the continental shelf, with 10Be concentrations suggesting retreat occurring during the mid- to later-Younger Dryas (YD). While this apparent retreat at face value presents a conundrum for ice sheet forcing (cold atmospheric forcing should drive ice sheet advance), the authors explore a variety of possible explanations for retreat, including melting of the ice front by warm subsurface ocean currents and the recent

suggestions that enhanced seasonality during the YD may obscure warmer summer temperatures in annually-average temperature reconstructions (Note: the short comment already posted by Richard Alley suggests that the authors may also consider the finer detail in the ice core data across the late-YD).

In addition, this dataset presents an important constraint on the timing of GrIS retreat from the continental shelves. The locations of these surface exposure dates are significantly proximal to the GrIS in comparison to the underwater moraines and grounding-zone wedges, which were previously inferred to be YD in age. This suggests that the ice sheet was significantly smaller during the mid- to late-YD than previously thought, which implications for estimates of ice volume evolution throughout the deglaciation.

Given the novel location of this chronology and the implications for ice sheet forcings during the Younger Dryas, I believe that this manuscript is worthy of publication. The authors provide a nice description of ice margin dynamics during the YD, and I appreciated reading their extensive survey of the literature throughout southern Greenland. However, I am concerned about the sampling of bedrock and the potential for inheritance. The authors work to address these concerns, but I believe the some of the conclusions may be overstated, given the possibility that many of the reported ages may still provide a surface exposure timing that is too old for the true timing of deglaciation.

Specific Comments

Inheritance – The existing "interactive comment" from Nicolas Young lays out many of the important concerns that I share on topic of inheritance. In many applications, bedrock samples are prone to inheritance due to insufficient subglacial erosion to reset the cosmogenic clock (Bierman et al., 1999, Geomorphology; Colgan et al., 2002, GSA Bulletin; Corbett et al., 2013, GSA Bulletin; Briner et al., 2016, GRL). And since each site presented in this paper only has 3 samples, it is difficult to ascertain outliers (either too-old or too-young). The two sites with boulder measurements (Ravns Storo and Sermiligarsuk) make this concern clear, with boulder measurements significantly

younger than the neighboring bedrock (although at Ravns Storo, 1 out of 3 bedrock samples does line up with the boulder sample). Additionally, each of these sites with boulder/bedrock pairings only have one boulder sample, which also precludes assessment of the efficacy of these boulder samples alone, even though they appear to be consistent with some of the neighboring 14C chronologies. I recognize that the authors state that boulders were generally not present, so I do not mean to suggest that more samples are necessary when more samples are not available. However, I think the conclusions about mid-YD retreat should not be overstated without acknowledging that some of the final exposure ages may still be too old (i.e. the bedrock ages provide a "maximum" age). For example, on line 221, the authors write "the results show that the ice sheet margin in the area WAS retreating on the inner shelf during YD". I suggest being careful with language of absolute causality here ("...in the area MAY HAVE BEEN retreating...") and throughout the paper.

Mid-YD retreat – Out of the 6 sites presented in this paper, it appears that only 3 suggest mid-YD retreat (Buksefjord, Avigaat, and Paamiut). In addition, the exposure age of Avigaat includes a rather large range of uncertainty that actually spans the entire YD, and therefore does not provide a robust constraint for before, during, or after YD. Again, coupled with concerns about inheritance, I continue to think that the conclusion about mid-YD retreat is overstated.

Lack of YD readvance (mentioned throughout, e.g. lines 246-248, 280-281) – Could you be more explicit on what types of evidence (or lack thereof) suggest that ice did not readvance during the initial YD? If the possibility of inheritance exists at some (or all) of these sites, couldn't the Fiskebanke moraines still potentially be YD in age, thus providing the evidence for a YD re-advance?

Evidence for warm-based ice - Of particular concern related to the topic of inheritance is whether or not ice was sufficiently erosive to reset the cosmogenic clock on the bedrock surface. In particular, previous studies have suggested there to be minimal glacial erosion at fjord mouths on Baffin due to ice thinning and spreading (Briner et al.,

2006, Geol. Soc. Am. Bull.). I wonder if the authors could provide further description of the sampled surfaces. For example, documented presence of striations and glacial polish would indicate basal sliding. Given the concerns about inheritance, being able to document warm-based ice conditions would help provide some indication that this landscape experienced "some amount" of glacial erosion prior to exposure. In addition, the evidence of striations and glacial polish would suggest minimal post-glacial erosion (which would bias exposure ages to be too-young; the authors assume zero erosion in their age calculation).

Line 131 – the youngest bedrock sample is considered to be a "minimum age". Given the potential for inheritance, I am not sure this is a true minimum age (as might be in the case in 14C in post-glacial lake sediments). I think the likelihood of inheritance in bedrock should suggest that even the youngest bedrock ages are a "maximum".

Technical Comments

Site Averages – I was unable to find any description of the averaging statistics the authors are employing for each of the sites. What is the form of averaging (error weighted or straight mean)? What is the joint uncertainty in the average ages (error-weighted sigma or standard error)?

Line 229-230 – This sentence is confusing. Consider rewording.

Line 319 – what is meant by the phrase "under the same token"?

Line 324 – "…the authors have chosen to use uncalibrated 14C ages from land…" Which authors are being cited here? The previous sentence cites two papers. Or are "the authors" referring to the writers of this manuscript?

---

## Referee Comment (RC2) · David Roberts (Referee) · 25 Jun 2020

General comments

This papers present some new deglacial ages (10Be) for the SW coast of Greenland. Some of those ages suggest that ice had retreated to the present coast prior to, or during the YD, though problems with inconsistent/inherited cosmogenic radionuclides make the construction of robust regional ice sheet history challenging. Many other deglacial age estimates along this coast (10Be and 14C) suggest deglaciation between 11.5 and 10.5 k, and an alternative approach (the incorporation and combination of the new chronological data with other local deglacial records) would have produced a different (younger) range of deglacial histories for the coast. Hence, the arguments

relating to YD ice extent would have been somewnat different to the conclusions made in the paper.

The discussion element of this paper provides a useful review of the possible extent of the GrIS pre and during the YD, and assesses some of the driving mechanisms (oceanic and atmospheric) that may have driven GrIS oscillation. However, the first part of the paper, the deglaciation of the SW sector of the GrIS, becomes divorced from the later 'review ' element of the paper, and given the problems with inheritance and the mismatch with other coastal deglacial ages this needs to be addressed.

Specific comments

Abstract: The abstract would benefit from more detail relating to the present study. Why is the GrIS margin being situated on the inner shelf unexpected? As is demonstrated later, the position of the GrIS margin during the YD is poorly constrained in Greenland, but ice has often been shown to be on the inner shelf/near coast during the YD.

Li 39: 'During the YD the GrIS in most areas had its margin on the shelf' …... this contradicts the abstract.

Li 40: What is the rationale behind choosing these six sites in the SW? What are the key aims of this paper - to provide a detailed analysis of these new sites, or to provide a review of the YD in Greenland? The paper is more focussed on the second of these aims and is unbalanced because of this.

Li 51: 'younger stratified aquatic sediments'….please provide details.

Li 60-64: Based on pre-existing work (10Be/14C) the retreat of the ice to the coast is fairly well constrained to 11.5 to 10.5ka. The new sites are effectively also at the coast. At best they will reinforce our knowledge of the timing ice retreat to the coast but how will they help with the YD question? Ideally, you really need offshore cores and 14C samples to answer the YD question.

Figure 1(b) does not provide enough detail on the location of the sites. Add a more

[Figure]

detailed map. Perhaps split the area north and south. Spatially (a few km's), many of these sites are at the coast and very close to pre-existing sites that have been dated. Do they provide the necessary lateral extent to effectively differentiate deglacial ages?

Li 130-132: 'therefore consider a spread of old ages as "inheritance outliers", while the mean of clustered younger ages gives the most reliable deglaciation age'. Inheritance does cause problems, not least because all the samples could be affected by inheritance and it cannot be quantified in any of the samples. Perhaps a more statistically robust approach to this (e.g. Jones et al. 2019 or Roberts et al., 2020 - Uncertainty weighted means/Chi-squared/extreme studentised deviation test) would provide an alternative framework for assessing the outliers?

Have the author also thought about combining their new data with pre-existing ages to provide more robust local datasets. It's a question of the lateral extent (spatial) and rate of retreat (temporal), but it is worth considering and could provide an alternative framework for deglaciation (to compare against).

Li 136- 148: Buksefjord - deglacial age of 12.3 ± 0.2 ka in mid-Younger Dryas (based on two ages) – this is much older than all other reported sites locally (10.7 to 11.4 ka). So, this could be inheritance, or deglaciation in the skaegaard (where is this?) indicates that the fjord glaciers lingered in their troughs while the adjacent coastal areas became ice free - the uncertainty makes it difficult to know which.

Li 149-160: Fiskenæsset deglaciated at 13.3± 0.5 ka – a robust set of samples that point to pre YD deglaciation. Local deglaciation previously reported at 10.6 - 10.5 ka. 'Even though these ages are minimum constraints for deglaciation, it is not likely that they postdate the deglaciation of the outer archipelago with 2000 years. This indicates that also here the major outlet glaciers reached the inner shelf, while adjacent areas had been ice free for some time'... Please explain this concept further for the benefit of the reader, as their does not seem to be any evidence presented to support this statement.

Li 162 – 169: Ravns Storø – a mixed set of ages with the two youngest providing an age of 11.7±0.2 of deglaciation (post YD).

Lin171- 177: Avigaat - A large spread of ages 13.7 ± 1.1, 12.0 ± 0.5 and 10.3 ± 2.5 ka giving an average of 12.0 ± 1.6. A local 14C age of provide an age 11.3 cal. Kyrs BP (possible mid YD deglaciation, but his is very poorly constrained).

Li 179 – 193: Paamuit – deglaciation at 12.2 ± 0.2 ka (robust set of ages) ice-margin retreating from the inner shelf in mid-Younger Dryas. This probably overlaps within error with the Kuanersoq age of 11.7 ka (please clarify), though 12.2 ka is older that other local deglacial ages (11.2 – 11.0 ka). 'we suggest that an ice stream in the Kuanersoq trough remained at the inner shelf while the adjacent coastal areas became ice free'. Based on the site descriptions provided and figure 1 it is very difficult for the reader to follow or substantiate this.

Li 195 – 204: Sermiligaarsuk deglaciation at 10.9 ±2.3 ka based on one date. Only one other local deglacial date (9.7 cal. ka BP) - post YD deglaciation

It is worth noting that all these sites are essentially at the present coast. None of them give a consistent pattern for the timing of deglaciation from the inner shelf to the present day coastline. So, it is very difficult to make inferences about the behaviour of the GrIS pre or during the YD. What would happen if these new ages where averaged with other local deglacial ages? It would give a very different picture.

Discussion

Li207-225: There is some evidence here to suggest ice withdrawing from the inner shelf pre or during the YD, but the 10Be ages are inconsistent. The arguments relating to ice steams sitting in the troughs later than the peripheral interstream areas along the coast makes glaciological sense, but is not really substantiated in any way in this paper.

Li227 – 248: This section provides a good overview the deglacial history of some other

sectors of the ice sheet during the YD.

Li 250- 282: Moraines on the outer to mid shelf (Hellefisk and Fiskebanke). This part of the paper provides a brief review of the possible age of the moraines on the shelf and concludes they pre date the YD and where formed in response to a range of climatic and non-climatic forcing factors. But this is stepping in to a different set of questions with respect to the behaviour of the GrIS and is becoming divorced from the original focus of the paper. These are mid to outer shelf moraine systems that formed pre YD. They are not directly related to the coastal deglacial story that form the basis of the paper.

Li 284 – 336: The third section of the discussion highlights a number of discrepancies with respect to the dating of GZW's on the continental shelf around Greenland. Those that have been dated (14C) often infer pre YD formation, but other several recent studies have speculated that many mid –shelf GZWs could be YD in age based on "climate-correlated" records. I think many researchers will agree that the second approach is flawed. This section is essentially a mini review of the dating of GZWS on the continental shelf, but it is only partially linked/relevant to start of this paper. I am not sure what this paper wants to be - a review paper?

Li 338 – 365: The last section (5.5 ) of the discussion provides a review of possible forcing mechanisms for deglaciation of the GrIS on the continental shelf during early deglaciation through to the YD. Ocean forcing and increased seasonality with respect to summer/winter air temperatures are discussed (cold, arid winters + increase in sea ice v warmer summers). This explains why ice was largely undergoing retreat pre YD and during the YD (despite the ice core records showing regional cooling during the YD). These are really important issues when it comes to understanding GrIS response to climate change, but again this discussion is largely divorced from the study at the start of this paper (deglaciation of the SW coast of Greenland).

---

## Author Comment (AC1) · 14 Aug 2020

Thank you for the comments. Below we address the points raised by Richard Alley. We are grateful for the interest shown in our results and the additional references. We agree that several of our records go back only to mid/late YD, and an early YD re-advance followed by retreat cannot be excluded in these areas, (which would still make the YD a period of net ice-margin recession). However, areas with existing chronologies spanning through the YD to Allerød times - notably the the Disko Bugt shelf in West Greenland and Kangerlussuaq trough in East Greenland, but also the Scoresby Sund region in East Greenland and southernmost Greenland - show no obvious change in ice margin behaviour over the Allerød/YD boundary. Furthermore, the evidence cited for YD re-advance/stillstand - moraines and grounding zone wedges on

the shelf - has been dated mainly from the assumption that they should represent cold periods in the ice core chronology, and therefore cannot really tell us about the ice margin/temperature relation. Only a local ice cap in North Greenland experienced an initial YD advance (Larsen et al., 2016). So, at least there is at present limited evidence for YD re-advance/prolonged stillstand of the ice sheet margin. On the other hand, the majority of evidence points to retreat during the YD period. All the above said, and given the apparent variability in YD ice margin behaviour between areas, an early YD re-advance followed by retreat is still a possibility. We hope that our contribution will inspire further work on this problem, which also has a bearing on the future.

---

## Author Comment (AC3) · 15 Aug 2020

Response to referee Davis Ullman. We are grateful for these comments and will address them below – and in a revised manuscript. Response to specific points:

Inheritance: In our response to Nicolas Young we have addressed some of the issues raised here. In short, we still feel that a cluster of ages, also in bedrock, should give a good estimate of the age for deglaciation at a site, but will discuss multiple interpretations in the text.

Ice retreat during YD: Our data suggest that the ice margin had retreated to the coastal sites by mid-YD (Buksefjord, Avigaat, Paamiut) or late-YD (Ravns Storø). The Fiskenæsset trough was largely deglaciated during the Allerød, whereas the age un-

certainty in Sermiligaarsuk is too large to be meaningful. In the revised version we will make sure not to overstate these conclusions.

Lack of YD readvance: We acknowledge that we need to be more clear that lack of evidence is not evidence for absense. The features previously interpreted as evidence for YD re-advance/long-lasting stillstand, such as moraines and grounding zone wedges on the shelf, are dated mainly by reference to the ice core temperature record, as noted in the reply to Richard Alley. We cannot exclude the possibility that the Fiskebanke moraine in our area could date to the early-YD. However, in the northern end of this moraine belt, the older Hellefisk moraine is considered to be deglacial and dated to mid-YD times (Hogan et al. 2016), implying that neither the Hellefisk nor the younger Fiskebanke moraine would reflect initial YD cooling.

Bedrock surface: We will add more information about the nature of the bedrock surface at our sample sites in the revised manuscript to underline that the area was subject to glacial erosion. See also response to Nicolas Young. However, it should be noted that the bedrock surfaces are affected by postglacial weathering. We have not accounted for that in the calculations as we cannot estimate the amount of erosion. If we could account for the postglacial weathering the 10Be ages would be slightly older.

Technical Comments: Site averages: Given the small number of data points from each site we have calculated the straight mean and standard deviation on the clusters from each site.

Specific comments to passages in the text: Thank you for reminding us. We agree with these suggestions and will reformulate and clarify these passages

---

## Author Comment (AC4) · 16 Aug 2020

We are grateful for the comprehensive comments from referee David Roberts, which we will address in a revised manuscript and in the notes below.

Incorporating other local records would have given a different deglacial history: We don't quite understand this comment. We believe that we have discussed all relevant records and their relationship with our data for each of our study sites. Balancing own results and review, the inheritance problem, and mismatch with coastal records: We discuss our new data in relation to new published data on the YD, both from land and shelf. Therefore, our paper provides a mixture of new results and review, which we are pleased to see that the other reviewers think is relevant and interesting. Thus, we prefer

[Figure]

to keep the current structure of the discussion, but will try to further include the new data into the discussion in the revised manuscript. We acknowledge the inheritance problem, as noted in our response to Nicolas Young, and will make adjustments in a revised manuscript. As to the mismatch with coastal records see notes below.

Specific comments Ice margin on inner shelf: what is unexpected is that it retreated during the YD. We have reformulated this sentence to make it more clear.

Rationale behind sites: The rationale of selecting the six sites on the inner shelf is to constrain the time when a contiguous ice margin on the shelf broke up and gave way to discrete ice streams in transverse troughs. We discuss our new data in relation to new published data on the YD, both from land and shelf.

Aquatic as opposed to glacial (which is widespread on the shelf to the north). We will specify the categories differentiated in the report by Roksandic (1979).

Retreat of the ice to the coast constrained to 11.5 to 10.5ka: As noted in comments to Richard Alley, there is large variation in the timing of deglaciation of the outer coast in Greenland, ranging from Allerød to Early Holocene, and the localities we sampled were under-explored. However, we agree that offshore work is highly needed to pin-point the timing of initial deglaciation after LGM and subsequent ice margin history until landfall, but that is beyond the scope of our contribution.

We will redraw Fig. 1b to show the location of dates in greater detail..

Statistical approach: If the uncertainty of the individual ages overlap we use them to calculate a mean age. If not, we identify them as outliers. Given the small sample size – 3 samples per site, we have not used more sophisticated methods to identify outliers, but we will keep this advice in mind in future work.

Combining the new data with pre-existing ages: We have compared our new data to existing data from nearby areas. However, all of the other sites are located tens of kilometers away (at a minimum) and it seemed unjustified to calculate a mean age. We

did not use our new data to calculate retreat rates between the outer coast and the present ice margin as we feel this would be a different story, and anyway it would be just two points.

Buksefjord and Fisknenæsset: the comments raised about these two sites are very similar to the comments raised by Nicolas Young. We have addressed them in our response to his comments. Fiskenæsset. ….. local deglaciation at 10.6 - 10.5 ka: These minimum-constraint ages are from a trough 50 km to the north - considered to be the very last to be vacated of ice in this part of Greenland, postdating the deglaciation of the adjacent areas with several millenia (Weidick, 1976). Considering the large variations in YD ice margin behaviour, even between neighbouring troughs, and the early deglaciation of the shelf to the south we do not find our deglaciation ages out of line with previous studies, but we will discuss these data, and the "mismatch" between our data and those from the coast in greater detail.

Avigaat: We agree this is very poorly constrained as stated in the manuscript (and shown by the large uncertainty in fig. 3).

Paamiut: As noted in the text, the previous 11.7 ka 10Be date from the outer Kuaner-soq trough dates thinning of the ice stream in the outer fjord, not deglaciation of the coast. Retreat from the shelf trough and into the fjord was dated by extrapolation by the authors to c. 11.0, so there are no overlapping ages.

Sermiligaarsuk based on one date: yes, we agree, this is not a robust deglaciation age. Averaging with other local deglacial ages, calculating retreat rates? We're not sure we completely understand this comment. True that our sample sites are just one point on several paleo-flowlines of the ice sheet, and we have a limited ability to gain knowledge on ice history before and after the timing of ice retreat at each of our sites.

ice steams sitting in the troughs later than the peripheral interstream areas: we need to go into more detail about this, with reference to previous studies, such as Roberts et al. 2009.

Moraines on the outer to mid shelf (Hellefisk and Fiskebanke)... not directly related to the coastal deglacial story that form the basis of the paper: We don't agree. We mention the moraines because it has been suggested frequently that they are related to YD readvance/stillstand. Recent studies – also disregarding ours – suggest that they are metachronous, and in some places indeed from YD, but deglacial. Therefore, we favor keeping this part in the revised manuscript.

dating of GZW's on the continental shelf. See comments above

"...discussion is largely divorced from the study...": We believe that this part of the discussion is important as it sums up the current knowledge about the climate forcing during YD and provides an explanation of the ice margin behavior during YD cooling. See also comment to Richard Alley.

We are grateful for this opportunity to discuss our results, and will work the comments into a revised manuscript.

————————————————

---

## Author Response (AR1)

**Comments from editor and reviewers/our response in interactive discussion/Action taken with line number**

Editor

 I echo one of the comments that asked for more detailed site maps (including bathymetry) to help readers assess the spatial relations between dated localities for different studies. Please also consider adding individual dates that make up the site means to Fig 3.

*Action*: We have made more detailed maps, including detailed bathymetry in Fig.1 and camel back diagrams for the uncertainties in each sample (Fig. 3)

Richard Alley

This is a fascinating paper, and the authors are to be congratulated. The possibility is important that the new data document oceanic effect of warm subsurface waters on grounding zones during the Younger Dryas. Please note, though, that retreat during the middle and late Younger Dryas (which the bulk of the ages from this paper support) is also consistent with control by surface temperature as recorded in ice cores. Many proxy records of temperature in Greenland, including the deep ice cores, show a relatively large, rapid cooling at the start of the Younger Dryas, then gradual warming (with some oscillations) through the Younger Dryas, and a sharp rise at the end. The simplest translation of that record into ice-marginal position assuming rapid response to temperature yields net retreat during the Younger Dryas (the modelling by Vacco et al., 2009, doi:10.1016/j.quascirev.20 08.04.018 provides examples of how this behaviour can occur). Young et al. (2019, 10.1130/G46058.1) found moraines from land-terminating glaciers dating from within the Younger Dryas in the Ahklun Mountains of southern Alaska, and citations therein point to similar behavior elsewhere. It is thus likely that additional data or models will be required to separate the influence of air temperature from ocean temperature.

Response

Thank you for the comments. We are grateful for the interest shown in our results and the additional references. We agree that several of our records go back only to mid/late YD, and an early YD re-advance followed by retreat cannot be excluded in these areas, (which would still make the YD a period of net ice-margin recession).

However, areas with existing chronologies spanning through the YD to Allerød times - notably the the Disko Bugt shelf in West Greenland and Kangerlussuaq trough in East Greenland, but also the Scoresby Sund region in East Greenland and southernmost Greenland - show no obvious change in ice margin behaviour over the Allerød/YD boundary. Furthermore, the evidence cited for YD re-advance/stillstand - moraines and grounding zone wedges on the shelf - has been dated only from the assumption that they should represent cold periods in the ice core chronology, and therefore cannot really tell us about the ice margin/temperature relation. Only a local ice cap in North Greenland experienced an initial YD advance (Larsen et al., 2016).

So, at least there is at present limited evidence for YD re-advance/prolonged stillstand of the ice sheet margin. On the other hand, the majority of evidence points to retreat during the YD period.

All the above said, and given the apparent variability in YD ice margin behaviour between areas, an early YD re-advance followed by retreat is still a possibility as it apparently happened some place around Greenland. We hope that our contribution, if it reaches publication, will inspire further work on this enigma.

*Action*: We discuss the probability of early advance followed by retreat, which could be the case in areas where the record only goes back to mid YD. 308

Nicolas Young

This manuscript by Funder et al presents a series of 10Be ages from coastal locations in southwest Greenland that are then used to infer ice-sheet history during the Younger Dryas. This is very much a chronology paper, and many of their arguments are heavily dependent on how one interprets the new 10Be ages. If this entire SW Greenland coastal chronology were based on 10Be ages from erratic boulders perched on bedrock, the author's interpretation of the dataset would be much easier to digest; however, this chronology is almost exclusively from bedrock which presents some challenges. The authors state that these sampling regions were largely devoid of suitable erratic boulders. Thats fine, you can't sample what is not there. But, what to make of standalone 10Be ages from bedrock surfaces can be tricky and I am not sure the authors here have really given this enough thought. The primary issue here is that the odds of encountering isotopic inheritance (producing ages that are too old) are higher in bedrock than in erratics, and the authors even acknowledge this. Often times inheritance presents itself as obvious outliers, but there are certainly cases where inheritance is much more subtle and even exists as 'uniform' amounts of inheritance. Again, I think the authors somewhat acknowledge this because they go through great lengths in section 4 to convince the reader that their preferred site by site coastal deglaciation ages are NOT influenced by inheritance even though in some cases their preferred ages of coastal deglaciation are significantly older than the ages of adjacent regions. Why does this matter? If you take their 10Be ages at face value, then it allows the possibility of the ice margin retreating onto land prior to the start of the Younger Dryas, which in turn the authors use to argue that a major Younger Dryas related oscillation did not occur out on the shelf since everything on the shelf would have to be pre-YD in age. However, a lot of this argument is resting on their 13.3 ka and 12.3 ka sites, Fiskenæsset and Bukesefjord. These ages are noticeably older than the constraints on deglaciation from adjacent areas. The authors take these ages at face value, but I think it is more likely that these suspiciously old sites have slight amount amounts of isotopic inheritance. If the 'true' age of deglaciation is in fact younger and more similar to the remainder of the deglaciation constraints from SW Greenland, the the ice margin is still somewhere out on the shelf during the Younger Dryas. Therefore you cannot rule out a major early or mid-YD oscillation of the ice margin. Considering all of the 10Be ages from Larsen 2014 and Winsor 2015, and all of the 14C ages from the region, the bulk of the deglaciation chronology around Nuuk (14C + 10Be) suggests deglaciation occurred 10.5 -10.7 ka, with a single older 14C age of around 11.4 ka. With that being the case, I think it is much harder to sell that the Fiskenæsset and Bukesefjord regions deglaciated several thousand years earlier. Note that the region between Fiskenæsset and Bukesefjord also deglaciated at the same time as the Nuuk region. Considering all of this evidence, coupled with bedrock only sampling, leads me to think that these 10Be ages are influenced by a bit of inheritance. To be sure, 10Be ages from bedrock can work, but it is typically only in select environments. The most obvious from the region that come to mind are Briner et al (2009; Nature Geoscience) from Baffin Island and Young et al (2013; Quat. Sci. Rev) from Disko Bugt. In both of these cases the 10Be chronologies rely pretty heavily on bedrock samples. Also in both cases the 10Be ages agree with the 14C constraints that exist. In particular, up in Disko Bugt the 14C and 10Be constraints are near identical. This consistency suggests that the bedrock-based 10Be ages are not influenced by inheritance. But what these sites have in common is that the bedrock is located is some of the most erosive environments on Earth where the chances of encountering inheritance should be minimal. On Baffin, samples are from bedrock knobs either right

in or adjacent to the Fiord trough (Sam Ford Fiord), and in Disko, these sites are directly adjacent to arguably the fastest glacier on Earth. In this manuscript, the authors are presenting 10Be ages from bedrock in coastal locations \*between\* former fast-flowing outlets. I wouldn't exactly consider this a recipe for success if you want to avoid inheritance in bedrock surfaces. Even sampling in highly erosive environments does not ensure inheritance-free samples. For example, Hughes et al (2012; Geology) sampled bedrock and erratics in Sermilik Fjord (Helheim) and present pretty convincing evidence that the bedrock samples are influenced by a small (only a few ka) and relatively uniform amount of inheritance. Co-author Briner himself demonstrated the same thing in Norway (2016; Geophysical Research Letters) where on the surface his 10Be ages from erratics are fairly uniform and likely come from what should be an erosive environment. But after considering other constraints, they concluded that their samples are likely influenced by a uniform baseline amount of inheritance equating to only a few ka. As we develop more and more of these 10be-based chronologies, its becoming clear that that small amounts of uniform inheritance exist and bedrock-based ages need to be treated with extra caution. I look at this dataset and the pre-existing chronology and highly suspect that many of the new 10Be ages in this manuscript have slight isotopic inheritance. And, I think the authors have significantly overreached in their interpretations on what the ice sheet may or may not have done during the YD based on their new 10Be ages. That is not to say these 10Be ages shouldn't be published; they are a valuable contribution that others will certainly build off. It is even possible that these ages are 'correct' but you would need to back the bedrock-only results with matching ages from erratics or any other independent constraint before you launch into sweeping interpretations about the ice margin during YD times.

Response

Thank you for the comments. We are grateful for the interest shown in our results and we will address the points below.

We agree that sampling erratic boulders perched on bedrock or on a moraine would have been ideal to minimise the potential problems with nuclide inheritance (which can plague bedrock samples in some, but not all, locations), but this was not possible at our sample sites. Few boulders were found, and thus our dataset largely consists of bedrock samples. Although not ideal, this dataset still has value, mainly because the low-lying coastal archipelago in our study area is the product of intense erosion by warm-based ice, probably during several ice ages and for the better part of the last Ice Age (Seidenkrantz et al., 2019). Therefore, the sample sites are at least not an obvious candidate for nuclide inheritance, much in the way that other parts of west Greenland in terrains of aerial scouring produce bedrock exposure ages with little-to-no evidence for inheritance (Young et al., 2013). However, it is clear from our data that some of the bedrock samples show signs of possible inheritance, recognized as variable ages with some well older than a cluster of others. We have addressed these indications of inheritance by using the youngest group of ages as the most likely deglaciation age of the area.

*Action:* We discuss the probability of uniform inheritance. "Deep" uniform inheritance is unlikely, but "shallow" uniform inheritance is a possibility. In both cases independent dating is required to control this potential error, which may be more widespread both in bedrock and boulders, than hitherto thought. 265ff

The two oldest of our sample sites (Buksefjord, average age of 12.3 ka and Fiskenæsset, average age of 13.3 ka) are especially suspicious to the reviewer because it would move the deglaciation of the inner shelf prior too or in early YD i.e. making a strong argument against the Fiskebanke moraines in this area as being not connected to the YD, as previously hypothesized (Funder et al., 2011). Perhaps also because these deglaciation ages are a little older than those from our other sites. On the other

hand, the ages from these two sites are internally consistent (low scatter) with only one obvious outlier at Buksefjord. The reviewer raises the possibility that all samples might be affected by small amount of uniform nuclide inheritance from long exposure durations combined with light glacial erosion during brief glacial occupations (cf. Briner et al., 2016). The reviewer points out that the deglaciation of Buksefjord is significantly older (by 1.6 kyr) than 10Be ages of boulders and 14C ages of marine molluscs from nearby sites around Buksefjord, and suggest that uniform inheritance may be the cause of this. Although we cannot rule out that inheritance is a possibility, we find it, as noted above, not likely. We favor our current interpretation that the coastal areas between fjords and troughs became ice free earlier than in the troughs, which were perhaps occupied later by lingering ice streams. This may apply also to our oldest site at Fiskenæsset, which implies that the ice margin was close to the coast in late Allerød times (13.3 ka). Here, there are no controlling data from nearby land. However, in the adjacent area to the south, the ice margin had already retreated from the shelf by the late YD (Sparrenbom et al. 2013; Levy et al., 2020), and farther south, the ice margin retreated on land already before the YD (Bennike et al., 2002; Levy et al., 2020). So, a deglaciation age of the coastline in the northern parts of ours study area in Allerød-early YD times would at least not be in conflict with data from neighbouring areas.

To sum up: NY raises several important issues concerning our dataset, especially about nuclide inheritance in our ages. We will modify our manuscript to discuss alternative interpretations of our ages, and then provide support for our favored interpretation.

*Action*: we discuss alternative interpretations, and loosen the conclusions. E.g. 194, 260ff

A few other minor things caught my eye as I read this:
1) It says you used version 3 of the CRONUS calculator for 10Be ages, and a production rate of 3.96. Version 3 uses an updated treatment of muon production, so the production rate needs to be recast with the same treatment of muon-based production. So if you used the online calculator, you would have to input all the Baffin Bay calibration data to make your own calibration, and then calculate the ages (there is an option to do this). In this case, the calculator will do the re-casting for you and the production rate is 4.04+-0.07; 3.96 is the number for the old scheme. Your ages should not change, but to avoid confusion, you should list the right PR. Or, you actually used version 2 of the CRONUS code, in which case that 3.96 number still applies (see Young et al., 2020; QSR)

Response

Thank you for pointing this out. We used Cronus version 3 with a production rate of 4.04+-0.07 i.e. the ages are correct. We will change this in the revision.

*Action*: has been done. (Table 1)

2) The authors spend a lot of time going around Greenland compiling constraints related to the YD ice margin position. Thus I was a bit surprised not to see a fairly robust constraint from near Sisimiut. Young et al (2020; QSR) present several 10Be ages from erratics directly at the coast and from moraine boulders also at the coast, that all date to ~11.6 ka. Therefore the ice margin was out on the shelf during the entirety of the YD.

We agree that the new evidence presented in Young et al. (2020) clearly demonstrates that the ice margin was out on the shelf during the YD – just as it was on the Disko shelf immediately to the north. The same is the case for SE Greenland with early Holocene dates along the coast (Dyke et al., 2018, Levy et al., 2020), and in NW Greenland (Søndergaard et al., 2020). However, in some areas around Greenland, ice DID retreat onto land prior to the YD (northernmost and southernmost

Greenland, Scoresby Sund). So it is not inconsistent that there might be other coastal areas in SW Greenland where there is a quite narrow continental shelf that ice pulled back to land during the late Allerød or early YD. We have incorporated these new references and discussion into the revised version of the manuscript.

**Action**: we have added NY's date as well as similar dates on the SE coast to emphasize that in some areas the ice margin stood on the shelf at the end of YD or early Preboreal, and there is no record of what happened to it during YD. 307, Fig. 4.

3) The authors also mention that they do not correct for uplift on 10Be ages because those effects are offset to some degree by atmospheric pressure variations, and they cite Young et al (2020; QSR). While this is true, a more robust thing you can say here is that the 10Be production rate calibration dataset likely underwent a similar amount of uplift to your known sites and therefore no correction is needed, the correction is essentially 'built in" in this case.

Response

We agree and we will incorporate this in the revised manuscript.

*Action:* has been done, 139

David Ullman
General Comments
This manuscript presents a new cosmogenic 10Be chronology for Greenland Ice Sheet (GrIS) retreat from an archipelago of outer islands off the southwest coast of Greenland. Surface exposure dates from these locations help to constrain the timing of retreat off the continental shelf, with 10Be concentrations suggesting retreat occurring during the mid- to later-Younger Dryas (YD). While this apparent retreat at face value presents a conundrum for ice sheet forcing (cold atmospheric forcing should drive ice sheet advance), the authors explore a variety of possible explanations for retreat, including melting of the ice front by warm subsurface ocean currents and the recent suggestions that enhanced seasonality during the YD may obscure warmer summer temperatures in annually-average temperature reconstructions (Note: the short comment already posted by Richard Alley suggests that the authors may also consider the finer detail in the ice core data across the late-YD).
In addition, this dataset presents an important constraint on the timing of GrIS retreat from the continental shelves. The locations of these surface exposure dates are significantly proximal to the GrIS in comparison to the underwater moraines and grounding- zone wedges, which were previously inferred to be YD in age. This suggests that the ice sheet was significantly smaller during the mid- to late-YD than previously thought, which implications for estimates of ice volume evolution throughout the deglaciation.

Given the novel location of this chronology and the implications for ice sheet forcings during the Younger Dryas, I believe that this manuscript is worthy of publication. The authors provide a nice description of ice margin dynamics during the YD, and I appreciated reading their extensive survey of the literature throughout southern Greenland. However, I am concerned about the sampling of bedrock and the potential for inheritance. The authors work to address these concerns, but I believe the some of the conclusions may be overstated, given the possibility that many of the reported ages may still provide a surface exposure timing that is too old for the true timing of deglaciation.

Response

Thank you for the comments. We will address the comments in the revised manuscript. See our response below.

Specific Comments

Inheritance – The existing "interactive comment" from Nicolas Young lays out many of the important concerns that I share on topic of inheritance. In many applications, bedrock samples are prone to inheritance due to insufficient subglacial erosion to reset the cosmogenic clock (Bierman et al., 1999, Geomorphology; Colgan et al., 2002, GSA Bulletin; Corbett et al., 2013, GSA Bulletin; Briner et al., 2016, GRL). And since each site presented in this paper only has 3 samples, it is difficult to ascertain outliers (either too-old or too-young). The two sites with boulder measurements (Ravns Storo and Sermiligarsuk) make this concern clear, with boulder measurements significantly younger than the neighboring bedrock (although at Ravns Storo, 1 out of 3 bedrock samples does line up with the boulder sample). Additionally, each of these sites with boulder/bedrock pairings only have one boulder sample, which also precludes assessment of the efficacy of these boulder samples alone, even though they appear to be consistent with some of the neighboring 14C chronologies. I recognize that the authors state that boulders were generally not present, so I do not mean to suggest that more samples are necessary when more samples are not available. However, I think the conclusions about mid-YD retreat should not be overstated without acknowledging that some of the final exposure ages may still be too old (i.e. the bedrock ages provide a "maximum" age). For example, on line 221, the authors write "the results show that the ice sheet margin in the area WAS retreating on the inner shelf during YD". I suggest being careful with language of absolute causality here (". . .in the area MAY HAVE BEEN retreating. . .") and throughout the paper.

Response

In our response to Nicolas Young we have addressed some of the issues raised here. In short, we still feel that a cluster of ages, also in bedrock, should give a good estimate of the age for deglaciation at a site, but will discuss multiple interpretations in the text

*Action*: has been done, see above.

Mid-YD retreat – Out of the 6 sites presented in this paper, it appears that only 3 suggest mid-YD retreat (Buksefjord, Avigaat, and Paamiut). In addition, the exposure age of Avigaat includes a rather large range of uncertainty that actually spans the entire YD, and therefore does not provide a robust constraint for before, during, or after YD. Again, coupled with concerns about inheritance, I continue to think that the conclusion about mid-YD retreat is overstated.

Response

Our data suggest that the ice margin had retreated to the coastal sites by mid-YD (Buksefjord, Avigaat, Paamiut) or late-YD (Ravns Storø). The Fiskenæsset trough was largely deglaciated during the Allerød, whereas the age uncertainty in Sermiligaarsuk is too large to be meaningful. In the revised version we will make sure not to overstate these conclusions.

Lack of YD readvance (mentioned throughout, e.g. lines 246-248, 280-281) – Could you be more explicit on what types of evidence (or lack thereof) suggest that ice did not readvance during the initial YD? If the possibility of inheritance exists at some (or all) of these sites, couldn't the Fiskebanke moraines still potentially be YD in age, thus providing the evidence for a YD re-advance?

Response

We acknowledge that we need to be more clear on this point and appreciate the comment. The features previously interpreted as evidence for YD re-advance/long-lasting stillstand, such as moraines and grounding zone wedges on the shelf, are dated only by reference to the ice core temperature record, as noted in the reply to Richard Alley. We cannot exclude the possibility that the Fiskebanke moraine in our area could date to the early-YD, except in the Fiskenæsset area where we favour the interpretation that our ages indicate an ice sheet margin close to the coast before the onset of YD. However, in the northern end of this moraine belt, the older Hellefisk moraine is considered to be deglacial and dated to mid-YD times (Hogan et al. 2016), implying that here the younger Fiskebanke moraine would not reflect initial YD cooling.

*Action*: We have avoided "lack of evidence" and shortened the moraine-discussion. Yes, the Fiskenæsset moraines could be YD - our Fiskenæsset dates can be interpreted both ways, and there is no other evidence available to date the moraines. We hope that someone will meet the challenge to go and date these moraines, which would be the most significant response to YD collong in Greenland. 331ff

Evidence for warm-based ice - Of particular concern related to the topic of inheritance is whether or not ice was sufficiently erosive to reset the cosmogenic clock on the bedrock surface. In particular, previous studies have suggested there to be minimal glacial erosion at fjord mouths on Baffin due to ice thinning and spreading (Briner et al., 2006, Geol. Soc. Am. Bull.). I wonder if the authors could provide further description of the sampled surfaces. For example, documented presence of striations and glacial polish would indicate basal sliding. Given the concerns about inheritance, being able to document warm-based ice conditions would help provide some indication that this landscape experienced "some amount" of glacial erosion prior to exposure. In addition, the evidence of striations and glacial polish would suggest minimal post-glacial erosion (which would bias exposure ages to be too-young; the authors assume zero erosion in their age calculation).

Response

We will add more information about the nature of the bedrock surface at our sample sites in the revised manuscript to underline that the area was subject to glacial erosion. See also response to NY. However, it should be noted that the bedrock surfaces are affected by postglacial weathering. We have not accounted for that in the calculations as we cannot estimate the amount of erosion. If we could account for the postglacial weathering the 10Be ages would be slightly older.

*Action*: has been done

Line 131 – the youngest bedrock sample is considered to be a "minimum age". Given the potential for inheritance, I am not sure this is a true minimum age (as might be in the case in 14C in post-glacial lake sediments). I think the likelihood of inheritance in bedrock should suggest that even the youngest bedrock ages are a "maximum".

Response

We agree and have changed that in the revised manuscript.

*Action*: changed 158, 237

Technical Comments
Site Averages – I was unable to find any description of the averaging statistics the authors are employing for each of the sites. What is the form of averaging (error weighted or straight mean)?

**Commented [NKL1]:** Svend, kan du tilføje den her?

What is the joint uncertainty in the average ages (error-weighted sigma or standard error)?

Response

Given the small number of data points from each site we have calculated the straight mean and standard deviation

> **Commented [NKL2]:** Svend, kan du tilføje hvordan det er beregnet.

Line 229-230 – This sentence is confusing. Consider rewording.

Response

We will rewrite the sentence.

*Action*: done
Line 319 – what is meant by the phrase "under the same token"?

Response

We will rewrite the sentence to make it clear.

*Action*: done

Line 324 – "...the authors have chosen to use uncalibrated 14C ages from land..." Which authors are being cited here? The previous sentence cites two papers. Or are "the authors" referring to the writers of this manuscript?

Response

We will rewrite the sentence and make it clear who we cite i.e. Arndt et al. 2017, Arndt 2018,

*Action*: done 363

David Roberts
General comments
This paper present some new deglacial ages (10Be) for the SW coast of Greenland. Some of those ages suggest that ice had retreated to the present coast prior to, or during the YD, though problems with inconsistent/inherited cosmogenic radionuclides make the construction of robust regional ice sheet history challenging. Many other deglacial age estimates along this coast (10Be and 14C) suggest deglaciation between 11.5 and 10.5 k, and an alternative approach (the incorporation and combination of the new chronological data with other local deglacial records) would have produced a different (younger) range of deglacial histories for the coast. Hence, the arguments relating to YD ice extent would have been somewhat different to the conclusions made in the paper.

Response
Thank you for the comments. We will address the comments in the revised manuscript. See our response below.

We don't quite understand the comment that we ought to have incorporated other local deglacial records, and this would have given a different deglacial history. We believe that we have discussed all relevant records and their relationship with our data for each of our study sites.

*Action*: as noted we don't see how other deglacial records could change our history.

The discussion element of this paper provides a useful review of the possible extent of the GrIS pre and during the YD, and assesses some of the driving mechanisms (oceanic and atmospheric) that may have driven GrIS oscillation. However, the first part of the paper, the deglaciation of the SW sector of the GrIS, becomes divorced from the later 'review ' element of the paper, and given the problems with inheritance and the mismatch with other coastal deglacial ages this needs to be addressed.

Response

We discuss our new data in relation to new published data on the YD, both from land and shelf. Therefore, our paper provides a mixture of new results and review, which we are pleased to see that the other reviewers think is relevant and interesting. Thus, we will keep the current structure of the discussion, but try to further include the new data into the discussion in the revised manuscript.

*Action*: as noted, we believe that our data need to be seen in a wider context, and have retained the review and discussion, although somewhat reduced.

Specific comments
Abstract: The abstract would benefit from more detail relating to the present study. Why is the GrIS margin being situated on the inner shelf unexpected? As is demonstrated later, the position of the GrIS margin during the YD is poorly constrained in Greenland, but ice has often been shown to be on the inner shelf/near coast during the YD.

Response

It is not unexpected that the ice margin is located on the inner shelf during YD. It is rather unexpected that it retreated during the YD. We have reformulated this sentence to make it more clear.

*Action*: unexpected is removed and areas where the ice margin stood on the shelf during YD have been added, Fig.4Glaciofluvial

Li 39: 'During the YD the GrIS in most areas had its margin on the shelf' ..... this contradicts the abstract.

Response

See comment above

Li 40: What is the rationale behind choosing these six sites in the SW? What are the key aims of this paper - to provide a detailed analysis of these new sites, or to provide a review of the YD in Greenland? The paper is more focussed on the second of these aims and is unbalanced because of this.

Response

The rationale of selecting the six sites in SW Greenland is to constrain the timing of deglaciation along the outermost coast, and these areas had little prior chronology. We specifically selected islands outside the main coastline to constrain the earliest deglaciation of the area. We discuss our new data in relation to new published data on the YD, both from land and shelf.

Li 51: 'younger stratified aquatic sediments'. . ..please provide details.

Response
Aquatic as opposed to glacial (which is widespread on the shelf to the north). We will specify the categories differentiated in the report by Roksandic (1979).
*Action*: 66

Li 60-64: Based on pre-existing work (10Be/14C) the retreat of the ice to the coast is fairly well constrained to 11.5 to 10.5ka. The new sites are effectively also at the coast. At best they will reinforce our knowledge of the timing ice retreat to the coast but how will they help with the YD question? Ideally, you really need offshore cores and 14C samples to answer the YD question.

Response

There is large variation in the timing of deglaciation of the outer coast in Greenland (see comments to Richard Alley) and the places we dated were under-explored. However, we agree that offshore work is highly needed to pin-point the timing of initial deglaciation after LGM and subsequent ice margin history until landfall, but that is beyond the scope of our contribution.

Figure 1(b) does not provide enough detail on the location of the sites. Add a more detailed map. Perhaps split the area north and south. Spatially (a few km's), many of these sites are at the coast and very close to pre-existing sites that have been dated. Do they provide the necessary lateral extent to effectively differentiate deglacial ages?

Response

We will make a new map to show all the details in the various sample sites
*Action*: done Fig. 1

Li 130-132: 'therefore consider a spread of old ages as "inheritance outliers", while the mean of clustered younger ages gives the most reliable deglaciation age'. Inheritance does cause problems, not least because all the samples could be affected by inheritance and it cannot be quantified in any of the samples. Perhaps a more statistically robust approach to this (e.g. Jones et al. 2019 or Roberts et al., 2020 - Uncertainty weighted means/Chi-squared/extreme studentised deviation test) would provide an al- ternative framework for assessing the outliers?

Response

If the uncertainty of the individual ages overlap we use them to calculate a mean age. If not, we identify them as outliers. Given the small sample size – 3 samples per site, we have not used more sophisticated methods to identify outliers.

*Action*: none

Have the author also thought about combining their new data with pre-existing ages to provide more robust local datasets. It's a question of the lateral extent (spatial) and rate of retreat (temporal), but it is worth considering and could provide an alternative framework for deglaciation (to compare against).

Response

We have compared our new data to existing data from nearby areas. However, all of the other sites are located tens of kilometers away (at a minimum) and it seemed unjustified to calculate a mean age. We did not use our new data to calculate retreat rates between the outer coast and the present ice margin as we feel this would be a different story, and anyway it would be just two points.

*Action*: none

Li 136- 148: Buksefjord - deglacial age of 12.3 ± 0.2 ka in mid-Younger Dryas (based on two ages) – this is much older than all other reported sites locally (10.7 to 11.4 ka). So, this could be inheritance, or deglaciation in the skaegaard (where is this?) indicates that the fjord glaciers lingered in their troughs while the adjacent coastal areas became ice free - the uncertainty makes it difficult to know which.

Response

The comments raised about the Buksefjord and Fiskenæsset are very similar to the comments raised by Nicolas Young. We have addressed them above.

Li 149-160: Fiskenæsset deglaciated at 13.3± 0.5 ka – a robust set of samples that point to pre YD deglaciation. Local deglaciation previously reported at 10.6 - 10.5 ka. 'Even though these ages are minimum constraints for deglaciation, it is not likely that they postdate the deglaciation of the outer archipelago with 2000 years. This indicates that also here the major outlet glaciers reached the inner shelf, while adjacent areas had been ice free for some time'. . . Please explain this concept further for the benefit of the reader, as their does not seem to be any evidence presented to support this statement. Li 162 – 169: Ravns Storø – a mixed set of ages with the two youngest providing an age of 11.7±0.2 of deglaciation (post YD).

Response

The minimum constraint ages of 10.6-10.5 are from a trough 50 km to the north – a trough which was considered to be the very last to be vacated of ice in this part of Greenland, postdating the deglaciation of the adjacent areas with several millenia (Weidick, 1976). Considering the large variations in YD ice margin behaviour, even between neighbouring troughs, and the early deglaciation of the shelf to the south we don't find our deglaciation ages out of line with previous studies, but see a need to discuss these data in greater detail.

*Action*: done 278

Lin171- 177: Avigaat - A large spread of ages 13.7 ± 1.1, 12.0 ± 0.5 and 10.3 ± 2.5 ka giving an average of 12.0 ± 1.6. A local 14C age of provide an age 11.3 cal. Kyrs BP (possible mid YD deglaciation, but his is very poorly constrained).

Response

We agree this is very poorly constrained as stated in the manuscript (and shown by the large uncertainty in fig. 3).
*Action*: none

Li 179 – 193: Paamiut – deglaciation at 12.2 ± 0.2 ka (robust set of ages) ice-margin retreating from the inner shelf in mid-Younger Dryas. This probably overlaps within error with the Kuanersoq age of 11.7 ka (please clarify), though 12.2 ka is older that other local deglacial ages (11.2 – 11.0 ka). 'we suggest that an ice stream in the Kuanersoq trough remained at the inner shelf while the adjacent coastal areas became ice free'. Based on the site descriptions provided and figure 1 it is very difficult for the reader to follow or substantiate this.

Response

As noted in the text, the previous 11.7 ka 10Be date from the outer Kuanersoq trough dates thinning of the ice stream, not deglaciation of the coast. Retreat from the trough on inner shelf is dated in prior work to c. 11.0 by extrapolation, so there are no overlapping ages. This area has been the subject of a large number of Quaternary studies over the years, each with very different purposes and aims. We have tried to sift out the information most relevant to our work.

*Action*: none

Li 195 – 204: Sermiligaarsuk deglaciation at 10.9 ±2.3 ka based on one date. Only one other local deglacial date (9.7 cal. ka BP) - post YD deglaciation.

It is worth noting that all these sites are essentially at the present coast. None of them give a consistent pattern for the timing of deglaciation from the inner shelf to the present day coastline. So, it is very difficult to make inferences about the behaviour of the GrIS pre or during the YD. What would happen if these new ages where averaged with other local deglacial ages? It would give a very different picture.

Response

We're not sure we completely understand this comment. True that our sample sites are just one point on several paleo-flowlines of the ice sheet, and we have a limited ability to gain knowledge on ice history before and after the timing of ice retreat at each of our sites.
*Action*: we have not averaged our ages with other ages from land

Discussion
Li207-225: There is some evidence here to suggest ice withdrawing from the inner shelf pre or during the YD, but the 10Be ages are inconsistent. The arguments relating to ice steams sitting in the troughs later than the peripheral interstream areas along the coast makes glaciological sense, but is not really substantiated in any way in this paper.

Response

We have elaborated more on this in the revised manuscript.

*Action*: done, see above

Li227 – 248: This section provides a good overview the deglacial history of some other sectors of the ice sheet during the YD.

Response

Thank you

Li 250- 282: Moraines on the outer to mid shelf (Hellefisk and Fiskebanke). This part of the paper provides a brief review of the possible age of the moraines on the shelf and concludes they pre date the YD and where formed in response to a range of climatic and non-climatic forcing factors. But this is stepping in to a different set of questions with respect to the behaviour of the GrIS and is becoming divorced from the original focus of the paper. These are mid to outer shelf moraine systems that formed pre YD. They are not directly related to the coastal deglacial story that form the basis of the paper.

Response

We don't agree. The new data partially constrain when the moraines on the shelf could have formed. Those moraines have not been directly dated (although their ages have been inferred plenty of times), so we argue that any new age information from the coastline also is relevant for the offshore shelf moraines. We mention the moraines because other recent studies have suggested that they are related to the YD and our data in the northern part of the study areas suggest otherwise. Therefore, we favor keeping this part in the revised manuscript.

*Action*: We have re-written a part of this discussion and added new evidence. 325ff

Li 284 – 336: The third section of the discussion highlights a number of discrepancies with respect to the dating of GZW's on the continental shelf around Greenland. Those that have been dated (14C) often infer pre YD formation, but other recent studies have speculated that many mid –shelf GZWs could be YD in age based on "climate-correlated" records. I think many researchers will agree that the second approach is flawed. This section is essentially a mini review of the dating of GZWS on the continental shelf, but it is only partially linked/relevant to start of this paper. I am not sure what this paper wants to be - a review paper?

Response

See comments above

Li 338 – 365: The last section (5.5 ) of the discussion provides a review of possible forcing mechanisms for deglaciation of the GrIS on the continental shelf during early deglaciation through to the YD. Ocean forcing and increased seasonality with respect to summer/winter air temperatures

are discussed (cold, arid winters + increase in sea ice v warmer summers). This explains why ice was largely undergoing retreat pre YD and during the YD (despite the ice core records showing regional cooling during the YD). These are really important issues when it comes to understanding GrIS response to climate change, but again this discussion is largely divorced from the study at the start of this paper (deglaciation of the SW coast of Greenland).

Response

We believe that this part of the discussion is important as it sums up the current knowledge about the climate forcing during YD and provides an explanation why the Greenland Ice Sheet responded to the YD cooling. See also comment to Richard Alley.

---

## Referee Report (RR1)

*Review of re-submission*

Funder et al present a revised manuscript based off a number of reviews/comments made on the initial submission. One of the primary concerns raised by not only me, but a few of the reviewers had to do with the issue of isotopic inheritance in bedrock samples. The main issue being that if you only have a few samples from one location, they are all from bedrock, and they yield consistent ages, one still cannot rule out the possibility that minor amounts of inheritance has influenced all the samples and the deglaciation age is slightly too old. This concept of minor "baseline" inheritance appears to exist within the literature. As it applies to this study, the concern is that if you take the 10Be ages at face value in a few locations then, yes, deglaciation of the outer coast could have happened early/mid Younger Dryas and then one can say the ice margin retreated through the Younger Dryas. If by chance, however, there is a small amount of inheritance and the "true" deglaciation age is younger (e.g., end of YD or later), then you cannot really say anything about ice-margin behavior during the YD because the ice margin was still in the ocean.

My primary criticism of version 1 was that the authors took the bedrock 10Be ages at face value and didn't really consider the alternative (inheritance), and then spun their dataset to argue for YD retreat at their sites with new 10Be ages. The argument in this revision is still the same, but I am pleased to see that the authors at least acknowledge the possibility of isotopic inheritance. For example, the topic of inheritance appears in the abstract, introduction, and at the start of section 4, no complaints on my end. I think really what we have here is a difference of interpretation. The authors default is to say these bedrock 10Be ages are "real" and then interpret those ages within the context of YD ice-margin behavior while, based on personal experience working in SW Greenland, I probably would have framed this study from the assumption that the bedrock samples might indeed be influenced by a slight amount of isotopic inheritance considering what else we know about regional deglaciation. Without a lot more measurements, there is no way say who is right here. Just a difference of scientific opinion, which should in no way hold up publication. The publication of the 10Be ages themselves is a contribution.

Moving beyond the inheritance issue, a few minor things I noticed that the authors should clean up:

- take another thorough look for typos and check where you have placed paragraph breaks.
- Might just be the version I have in front of me, but double check the photo resolution for images in Figure 2.
- I still advocate for up in the Sisimiut region, based on the recent 10Be ages from the coast, you can conclude 1) ice margin was out in the ocean until ~11.6 ka, and 2) the Fiskebanke moraines have to be >11.6 ka (Young et al, 2020). As it is, they place a line on a figure marking the "Preboreal" ice-margin position in the region, but I think that

glosses over the level of detail we actually know about the ice-margin history in that region based off recent 10Be ages.

- On Fig.4, shouldn't there also be a blue arrow in the Disko region as well since there is a well-constrained ice-margin advance in the early/middle YD, followed by retreat through the remained of the YD. Maybe touching upon Richard's point, but interpreting Greenland ice cores as a direct proxy for ice-margin behavior, one might expect an early/mid YD advance followed by retreat….which is what the chronology implies in Disko Bugt.

- Lastly, I encourage these authors to place their 10Be sample information in the ICE-D database (http://ice-d.org/). I personally think the community should be embracing this database as its more tailored towards the needs of cosmogenic isotope users versus some of the more generic repositories, and I think it is pretty user friendly. The authors cannot upload their dataset themselves, but if they contact me, I'd be happy to do it.

---

## Author Response (AR2)

Response to comments from referee Nicolas Young

We are grateful for NA's valuable and involved comments on both the first and second edition of our manuscript. As can be seen, the comments from NA and other referees have led to substantial changes in the revised ms. A somewhat depressing experience we have gained from this study is the role of $^{10}$Be inheritance in the bedrock surface, even in areas which have suffered intense glacial erosion. However, we still believe that the "cluster method" is viable for bedrock dating, but it requires many more dates from each site, which in most cases will probably be prohibitive for its application.

Specific comments:

Ad 1) "…The ice margin history in the Sisimiut area…glossed over…": We don't quite understand this comment. We are of course aware of the excellent recent study by NA and co-authors on the Holocene deglaciation history in this region, and as a response to the NA's comment on our first edition, with reference to this work, we added a Preboreal ice margin at the coast to signal that the YD ice margin here was further out on the shelf, implying (ad 2) that the Fiskebanke moraine here is older than Preboreal (which has never been contested). However, where the ice margin was, its behaviour during the YD climate oscillation, and the age of the Fiskebanke moraine remains unknown (as was also the case for the SE Greenland coast, and probably also other areas without a record).

Fig. 4 "…The well constrained ice margin readvance in Disko Bugt…": Yes, we have added a blue arrow on the Disko Bugt shelf, and give in the text some more details of this dramatic and enigmatic advance/retreat event. (We must admit that we find the original authors' arguments for a glacio-dynamic cause for this singular event quite convincing, as detailed in the new version).

As to the typos and figures: a number have been corrected, and the figures were designed for easy handling in e-mails. They will be better in a published edition. We will of course gladly submit our data to the database.

We have been glad to see that our small contribution to the YD problems was able to raise discussion at a high level. We would be even more glad, if it could be an inspiration to climate-independent dating of some of the features on the shelf (moraines, GZWs, and other), which have been attributed to YD glacier behaviour on climate grounds only. Understanding how the Greenland ice sheet reacted to climatic change in those turbulent times is essential for evaluating what is now about to happen.